# A Narrative Review: Immunometabolic Interactions of Host–Gut Microbiota and Botanical Active Ingredients in Gastrointestinal Cancers

**DOI:** 10.3390/ijms25169096

**Published:** 2024-08-22

**Authors:** Shanlan Li, Wuwen Feng, Jiaqi Wu, Herong Cui, Yiting Wang, Tianzhen Liang, Jin An, Wanling Chen, Zhuoqian Guo, Haimin Lei

**Affiliations:** 1School of Chinese Pharmacy, Beijing University of Chinese Medicine, Beijing 102488, China; 15901293873@163.com (S.L.); wjq020719@163.com (J.W.); 15684638965@163.com (Y.W.); polarisa1225@163.com (T.L.); 18810602247@163.com (J.A.); cwl6948@163.com (W.C.); gzq18332368157@163.com (Z.G.); 2School of Pharmacy, Chengdu University of Traditional Chinese Medicine, Chengdu 610075, China; fengwuwen@cdutcm.edu.cn

**Keywords:** gastrointestinal cancers, gut microbiota, oncometabolites, host–gut microbiota interaction, immunometabolism

## Abstract

The gastrointestinal tract is where the majority of gut microbiota settles; therefore, the composition of the gut microbiota and the changes in metabolites, as well as their modulatory effects on the immune system, have a very important impact on the development of gastrointestinal diseases. The purpose of this article was to review the role of the gut microbiota in the host environment and immunometabolic system and to summarize the beneficial effects of botanical active ingredients on gastrointestinal cancer, so as to provide prospective insights for the prevention and treatment of gastrointestinal diseases. A literature search was performed on the PubMed database with the keywords “gastrointestinal cancer”, “gut microbiota”, “immunometabolism”, “SCFAs”, “bile acids”, “polyamines”, “tryptophan”, “bacteriocins”, “immune cells”, “energy metabolism”, “polyphenols”, “polysaccharides”, “alkaloids”, and “triterpenes”. The changes in the composition of the gut microbiota influenced gastrointestinal disorders, whereas their metabolites, such as SCFAs, bacteriocins, and botanical metabolites, could impede gastrointestinal cancers and polyamine-, tryptophan-, and bile acid-induced carcinogenic mechanisms. GPRCs, HDACs, FXRs, and AHRs were important receptor signals for the gut microbial metabolites in influencing the development of gastrointestinal cancer. Botanical active ingredients exerted positive effects on gastrointestinal cancer by influencing the composition of gut microbes and modulating immune metabolism. Gastrointestinal cancer could be ameliorated by altering the gut microbial environment, administering botanical active ingredients for treatment, and stimulating or blocking the immune metabolism signaling molecules. Despite extensive and growing research on the microbiota, it appeared to represent more of an indicator of the gut health status associated with adequate fiber intake than an autonomous causative factor in the prevention of gastrointestinal diseases. This study detailed the pathogenesis of gastrointestinal cancers and the botanical active ingredients used for their treatment in the hope of providing inspiration for research into simpler, safer, and more effective treatment pathways or therapeutic agents in the field.

## 1. Introduction

Gastrointestinal cancers cover a wide range of diseases that affect various organs, such as the esophagus, stomach, liver, pancreas, colon, and rectum, causing a significant health burden worldwide [1]. In absolute numbers, pre-pandemic data estimated approximately 5 million new cases of gastrointestinal cancer in 2018, with more than 3 million associated deaths [2]. For example, *Helicobacter pylori* (HP) causes genetic instability and chronic gastric mucosal inflammation, subsequently leading to gastric tumorigenesis [3,4]. Some pathogens, such as *B. fragilis*, *F. nucleatum*, and *P. anaerobius*, may, via a variety of pathways, including the initiation of inflammation, the attachment of fine gut microbes to host cells, and the generation of toxins, contribute to the progress of colon cancer [5]. Additionally, some pathogenic gut microbes can interact directly with host cells to cause pro-inflammatory reactions and DNA damage, resulting in cancer [6]. The gut microbiota has a significant impact on regulating tumorigenesis and the progression and response to antitumor therapy [7].

The largest number of gut microbes, viruses, and fungi in our bodies can be found in the gut microbiota, primarily located in the colon [8]. The main mechanism by which the gut microbiota interacts with the host is through metabolites, which are the tiny molecules created as the intermediate or final byproducts of microbial metabolism. These can come directly from gut microbes, host molecules, or from the metabolism of dietary substrates by gut microbes [9]. These microbial metabolites send signals that affect immunological development, immune homeostasis, energy metabolism, and mucosal integrity maintenance [10,11]. In particular, they have a substantial effect on gastrointestinal cancer. These metabolites promote colorectal cancer (CRC) by increasing DNA damage and cancer cell proliferation, enhancing tumor invasion and metastasis, decreasing intercellular adhesion, and promoting genomic instability. They also inhibit the growth of CRC by assembling tight junctions, preventing inflammation, promoting the apoptosis of cancer cells, and protecting the colonic epithelial barrier [12]. Therefore, an improved comprehension of the gut microbiota that are involved in the biosynthesis of the metabolites associated with gastrointestinal cancers will greatly contribute to gut health management, especially for gastrointestinal cancer prevention.

The gut microbiota community is often exposed to potential health problems, such as inflammation and infection. To protect the host, the immune system maintains a cooperative association with the host’s gut microbiomes. On the other hand, to maintain the immune homeostasis of the host, the gut flora often play a non-negligible immunomodulatory function [13]. Immunometabolism has recently emerged as a potential therapeutic method for regulating inflammation by shifting inflammatory immune cells towards the anti-inflammatory spectrum by manipulating cellular metabolism. Immune cell metabolism is strictly controlled to preserve immunological homeostasis and functional specialization [14]. Modern scientific tools, including histopathology, genomics, immunoprofiling, single-cell transcriptomics, T cell receptor (TCR) characterization, neoantigen prediction, and in vitro T cell manipulation, have been used to study the relationship between immunity and the gut microenvironment [15]. Gut microbes play a substantial role in controlling the metabolic profile and functional responses of immune cells, owing to their extensive interactions with metabolites, intestinal epithelial cells, mucosal immune cells, and gut microorganisms. Accordingly, new therapeutic approaches for autoimmune diseases, inflammation, and cancer might be possible by developing methods that target the metabolic pathways of mucosal immune cells by modifying gut microbial metabolism [14]. The studies discussed in this review article may be the key to gaining a deeper understanding of the underlying mechanisms of response and resistance to these strategies and to comprehending the complexity of the immune environment in these gastrointestinal cancers.

## 2. Methods

This narrative review conforms to the Scale for the Assessment of Narrative Review Articles (SANRA) guidelines. As of March 2024, an electronic literature search was conducted on the PubMed database and CNKI using the Medical Subject Heading (MeSH) terms “gastrointestinal cancer”, “gut microbiota”, “immunometabolism”, “SCFAs”, “bile acids”, “polyamines”, “tryptophan”, “bacteriocins”, “immune cells”, “energy metabolism”, “polyphenols”, “polysaccharides”, “alkaloids”, and “triterpenes”. We considered the original studies, reviews, and meta-analyses obtained by searching for the above terms, while letters to the editor, editorials, expert opinions, and conference proceedings were excluded. The titles and abstracts of the search results were screened. The potentially relevant full manuscripts were read. The References Section lists the papers that were reviewed.

## 3. Gut Microbiota and Metabolites in Gastrointestinal Cancer

### 3.1. Short-Chain Fatty Acids (SCFAs)

Short-chain fatty acids (SCFAs) are the main products of gut microbial metabolism in the human colon, primarily coming from dietary carbohydrates like non-starch polysaccharides, oligosaccharides, lignans, and similar polysaccharides that are fermented by intestinal microorganisms without oxygen [16]. SCFAs mainly consist of butyrate, propionate, and acetate [17]. These SCFAs are an energy source for the colonocytes and perform an essential function in glucose metabolism and the regulation of cholesterol synthesis [18].

Among SCFAs, butyrate has been investigated the most and is mainly produced by the phylum *Firmicutes* (Table 1). One of the colon’s main energy sources is butyrate, and the prevention and inhibition of colon cancer are some of the beneficial effects of butyrate on human gut health [19]. Rodent models have shown that a higher butyrate supply lowers the risk of colon cancers caused by carcinogens, in part by inducing apoptosis [20,21]. Apoptosis is a very important event in the early stages of colorectal cancer development. This involves the removal of the genomically unstable cells that arise during tumorigenesis [22] and the deletion of the cells that have suffered DNA damage due to genotoxic agents such as carcinogens [23]. Enhanced apoptosis during the initiation events increases the elimination of mutant cells that might otherwise progress to malignancy [24] and defective apoptotic responses that are associated with an increased risk. It has also been demonstrated that this dose-dependent effect of butyrate to decrease colonic epithelial DNA damage and increase the rate of apoptotic deletion in colon cells is not associated with increased cell proliferation. Butyrate might accomplish this by inhibiting histone deacetylases. By inhibiting histone deacetylases (HDACs) and altering chromatin, butyrate induces the expression of the target genes that regulate the proteins involved in apoptosis, as well as cell cycle regulation and DNA repair [25,26]. Butyrate has also been shown to induce apoptosis through the activation of G protein-coupled receptor 43 (GPR43), which is frequently lost in colon cancer cells and is usually accompanied by the silencing of its genes encoding various receptors and transporter proteins involved in butyric acid-induced apoptosis in CRC cell lines [27,28]. For example, the butyrate transporter protein SLC5A8 gene is usually epigenetically silenced due to aberrant methylation in early human colon tumors. Butyrate supplementation promoted butyrate-induced apoptosis when it restored its expression [29]. In in vitro experiments, butyrate inhibited the proliferation of colon cancer cell lines and induced cell cycle arrest via multiple molecular pathways [30]. For example, in HT-29 human colon adenocarcinoma cells, butyrate blocked the cell cycle and inhibited cell proliferation by decreasing the expression of cdk2-stimulated cell cycle protein D and p21/WAF1/CIP1 [31]. In the CRC cell lines, butyrate downregulated p53 mRNA and protein expression and also directly increased the expression of the p53 target genes (e.g., p21WAF1, p27, and cell cycle protein-dependent kinases) to induce cell cycle arrest [32,33]. Wnt/β-catenin was a typical pathway induced by CRC, and undegraded β-catenin accumulated and freely translocated to the nucleus, thereby promoting the transcription of the genes involved in cell cycle regulation [34]. Butyrate promoted the translocation of β-catenin to autophagosomes and autolysosomes for degradation and the subsequent inhibition of CRC cell proliferation (Figure 1) [35]. The capacity of butyrate to influence cells has primarily been linked to its inhibitory action on HDAC, which causes histones to become hyperacetylated and increases the accessibility of transcription factors to nucleosomal DNA, thereby epigenetically controlling gene expression [36]. Butyrate might also target other intracellular targets, such as DNA methylation changes, the selective suppression of histone phosphorylation, the hyperacetylation of non-histone proteins, and changes in intracellular kinase signaling [37,38,39]. This diversity of effects might be the base of the ability of butyrate to regulate gene expression and affect the key regulators of apoptosis and the cell cycle [40]. Additionally, butyrate acted as an agonist for a number of G protein-coupled receptors (GPRs), such as Olfr78, GPR109A, GPR43, and GPR41 [41]. When activated in dendritic cells, GPRs increased the levels of the pro- and anti-inflammatory cytokines interleukin 17 (IL-17) and interleukin 10 (IL-10) and triggered the transformation of initial CD4+ T cells into inhibitory Tregs. Macia L reported that in some animal experimental models, such as the DSS-induced Gpr43^−^/^−^, GPR109a^−^/^−^ mouse colitis, and AOM/DSS-induced GPR176^−^/^−^ mouse CRC models, the deletion of GPR accelerated the development of colonic inflammation and colorectal cancer [42,43,44]. In contrast to the orthodox intestinal mucosa, the lack of GPR43 expression in the tissues of colorectal adenocarcinoma resulted in decreased expression in colorectal hyperplasia and benign colorectal disorders, including polyps [27]. GPR109A expression was dysregulated in colorectal cancers because of DNA methylation, and the receptor could be activated by butyrate in CRC cell lines, which eliminates Nuclear factor kappa-B (NF-κB) activation and induces apoptosis [45]. From a signaling pathway perspective, butyrate could halt colorectal cancer cell growth by inhibiting NF-κB and Wnt/β-catenin signaling. In colon cancer cells from humans, butyrate suppressed NF-κB activity by upregulating the transcription and activity of peroxisome proliferator-activated receptor γ (PPARγ). It also blocked NF-κB p65 signaling via a mechanism that is dependent on GPR109A [46,47]. Additionally, butyrate decreased the transcription of pro-inflammatory genes, such as NF-κB, in the primary epithelial cells of the human small intestine and colon, as well as the nuclear translocation of NF-κB and DNA binding in colon cancer cells [48,49]. Aryl hydrocarbon receptor (AHR) is a ligand-dependent transcription factor that, by restraining the Wnt/β-catenin signaling pathway and epithelial differentiation, regulates stem cell proliferation and preserves the integrity of the intestinal barrier when activated [50]. In addition, AHR activation prevents epithelial barrier dysfunction by upregulating Notch1 signaling and maintaining tight junction expression [51]. To the authors’ knowledge, the main reports on the mechanism of action of butyrate in colorectal cancer have mainly focused on the cellular level, but the signaling of the associated receptors has not been investigated at sufficient depth. Thus, further experiments are needed to reveal this topic.

The primary producers of propionate are *Phascolarctobacterium succinatutens*, *Bacteroides* spp., *Veillonella* spp., *Megasphaeraelsdenii*, *Coprococcuscatus*, *Dialister* spp., *Ruminococcusobeum*, *Roseburia inulinivorans*, and *Salmonella* spp. [52]. Protein arginine methyltransferase (PRMT1) is overexpressed early in CRC and contributes to the malignant characteristics of CRC progression. In HCT116 cells, propionate downregulates PRMT1, and PRMT1 decreases via siPRMT1 therapy, causing apoptosis by preventing phospho-p70S6 kinase. However, the mechanisms governing the relationship between propionate and PRMT1 need further investigation [53].

Acetate is primarily produced by *Akkermansia muciniphila*, *Anaplasma* spp., *Clostridium* spp., *Bifidobacterium* spp., *Prevotella* spp., *Rumenococcus* spp., *Streptococcus* spp., *Broutonella* spp., and *Putrefaciens* spp. [54,55,56]. Reports on the association between acetate and gastrointestinal cancer are relatively few, and studies have been limited to cellular experiments. Sodium acetate enhanced cell differentiation and programmed cell death; stimulated the release of histone D; caused lysosomal membrane permeability; and impeded the proliferation of CRC cell lines [57]. Its anticancer properties stem from its ability to initiate the death or necrosis of CRC cells via the mitochondria [58]. In addition, sodium acetate caused gastric adenocarcinoma cell growth in a dose-dependent manner [57].

### 3.2. Bile Acids

Bile acids (BAs) are cholesterol-producing steroid molecules found in hepatocytes [59]. BAs are reabsorbed in the small intestine to the ileum via BA transporter proteins after uncoupling by microbial bile salt hydrolase (BSH) [60]. Unresorbed BAs serve as a metabolic substrate for microorganisms and are converted to secondary bile acids [61]. *Bacteroides*, *Bifidobacterium*, *Eubacterium*, *Peptostreptococcus*, *Ruminococcus*, *Propionibacterium*, *Lactobacillus*, *Clostridium*, *Streptococcus*, *Escherichia*, and the archaeal genus *Methanobrevibacter* are common genera that have been identified for their involvement in bile acid metabolism [61]. BAs can directly affect the gut flora in both favorable and negative ways [62]. In the event of antibiotic- or toxin-induced dysbiosis, a primary BA such as taurocholate can stimulate spore germination, which might occur in a latent and non-toxic form and might serve as a homing signal for the gut microbiota [63]. Additionally, primary BAs might act as barriers that prevent harmful Gram-negative bacteria from proliferating in the small intestine [64]. A recent investigation of cirrhotic individuals with impaired bile acid biosynthesis revealed a positive correlation between a drop in the chenodeoxycholic acid (CDCA) densities and fecal densities of secondary BAs and an increase in potentially dangerous *Enterobacteriaceae*. This might be attributed to a decrease in the typical resident population of *Clostridia* capable of 7α-dehydroxylation (i.e., *Clostridium cluster XVIa*) [64]. Similarly, gut microorganisms affect BAs. The microbes in the ileum, such as *Lactobacillus*, *Staphylococcus*, *Streptococcus*, and *Veillonella*, usually cause the bile acid dehydroxylation of glycine or taurine via hydrolytic enzymes and oxidized hydroxyl radicals [65]. In the distal ileum, most BAs (95%) are reabsorbed and recirculated through the liver [65,66]. However, minute quantities proceed to the colon, where they function as substrates for many of the biotransformation processes carried out by gut microbes [65].

BAs are metabolized by gut microbes into secondary BAs, such as deoxycholic acid (DCA) and lithocholic acid (LCA), which have been suspected to produce carcinogenic or co-carcinogenic compounds [67,68]. For instance, the fecal bile acid levels were higher in cancer patients than those in healthy individuals or patients with other illnesses [68]. Additionally, the excrement of patients with colon cancer had higher levels of *Clostridium difficile*, indicating a connection between the bacterium and the illness [68]. In contrast, epidemiological data found no increase in gastrointestinal cancers in patients with the partial removal of the ileum, whereas the associated risk increased in patients with ileus, which might be related to the bile acid load of the colon [69]. These data imply that regulating the gut flora and bile acid metabolism could be essential for the prevention of a wide range of disorders in humans.

The pro-tumorigenic mechanisms of BA are mainly related to β-catenin and Wnt signaling. The effects of DCA and LCA on colon cancer cell lines have been demonstrated to include the stimulation of hyperproliferation, DNA fragmentation, and oxidative damage; the upregulation of cell cycle protein D1, which is involved in cell cycle progression through β-catenin cell signaling; the degradation of the tumor suppressor p53 (Figure 1); and the promotion of resistance to apoptosis [70,71,72,73,74]. In CRC rodents, DCA was utilized to cause low-grade intestinal inflammation; induce crypt dysplasia and intestinal carcinogenesis; improve epithelial Wnt signaling, as demonstrated by the accumulation of β-linker proteins and cytosolic protein D1; and compromise the integrity of the epithelial barrier [75,76,77].

### 3.3. Polyamine

In mammals, putrescine, spermidine, and spermine are produced sequentially through the catabolism of the amino acids L-arginine and L-methionine [78,79]. Putrescine is produced via the L-arginine water-l-ornithine/urea-ornithine decarboxylase (ODC) and spermidine–guanidine–polyamine pathways [80,81,82,83], after which spermine synthase (SMS) and spermine synthase (SRM) sequentially convert putrescine to spermidine and spermine [79,84]. Polyamine metabolism is crucial for cancer cell growth and normal cell proliferation. Prior research has demonstrated a connection between polyamine metabolism, gastrointestinal epithelial cell infection, and an elevated risk of HP-mediated gastric cancer [85,86]. The enzyme spermine oxidase (SMOX) is essential to the gastrointestinal system’s polyamine metabolism, efficiently regulating the levels of spermidine and spermine [85]. SMOX can cause DNA damage, which, in turn, can lead to gastric cancer (GC) [87,88]. Moreover, the mediator SMOX activates the oncogenic signaling pathway of the β-linker protein in HP stimulated gastric epithelial cells [89]. More than 80% of sporadic colorectal cancers are a consequence of adenomatous polyposis coli (APC) mutations [90], and the Wnt cascade response is largely mediated by the APC oncogene. The stimulation of the Wnt signaling pathway also upregulates Myelocytomatosis oncogene (MYC) [91], a transcriptional activator of ODC [92]. These mutations and dysregulated Wnt signaling eventually result in an increase in ODC expression, polyamine pools, and MYC activity [93]. Kirsten rat sarcoma viral oncogene (KRAS) has also been suggested to play a central role in both the early phases of colorectal cancer’s malignant development and progressive metastatic illness. Mutant KRAS upregulated ODC transcripts and downregulated SSAT transcripts, resulting in an increase in polyamine biosynthesis and a reduction in polyamine catabolism [94]. In addition, the endocytic process regulated by niche proteins allows cells to ingest polyamines. Niche protein-1 is phosphorylated as a result of KRAS activation, which significantly improves endocytosis and raises the amount of polyamines that are absorbed by cells [95]. Elevated polyamine levels promote the catalysis of eukaryotic initiation factor 5A (eIF5A), which triggers MYC biosynthesis and initiates a positive feedback loop that elevates CRC progression [96,97]. The gut microbiota is also a major contributor to polyamine production in the gut [98]. According to Sugiyama et al., the generation of polyamines from new polyamine biosynthesis and transporter proteins is influenced by the dominant gut microbiota [99]. *Enterococcus faecalis* can metabolize guanidinium to putrescine via the guanidinium deaminase pathway and develop pH resistance to colonize the intestinal ecological niche [100]. *Campylobacter jejuni* synthesizes spermidine via an alternative carboxyspermidine pathway [101]. *Bifidobacterium animalis* and *Bacillus lactic* subspecies are also involved in the metabolic processes that increase the polyamine putrescine levels [102,103]. In addition, Chevalier et al.’s macrogenomic analysis of ovariectomized mice revealed that exposure to warm environments increased the abundance of the genera that produced polyamines, such as *Bacteroides*, *Alitsipes*, and *Akkermansia muciniphila*, and decreased the abundance of the genera that degraded polyamines, such as *Muribaculaceae* and *Lachnospirae* expansion [104]. In obese mice, the alterations in polyamine levels, such as elevated spermidine levels, impacted the makeup and functionality of the gut microbiota. Spermidine exerted a gut microbiota-dependent anti-obesity effect via the expansion of *Lachnospirae NK4A136*, which improved the gut barrier function [89]. Spermidine also directly affected the synthesis of colistin toxin in *Esherichia coli* strains, suggesting a role for spermidine in microbial pathogenicity and carcinogenesis [105].

### 3.4. Tryptophan Metabolites

Tryptophan metabolites are derived from proteolytic metabolism and include indole, indole-3-acetic acid, indole-3-aldehyde, and indole-3-propionic acid. This catabolic step is mediated by gut microbes, and different microbial species have different catalytic enzymes, some of which cooperate with each other to produce tryptophan metabolites. For example, indole is produced by members of the *Firmicutes* phylum, such as *Clostridium limosum*, *Enterobacter aerogenes*, *C. tetani*, *C. bifermentans*, *C. lentoputrescens*, and *C. melanomenatum*, and some members of the *Bacteroidetes*, *Fusobacteria*, and *Proteobacteria* phyla, whereas indole-3-aldehyde (IAID) is produced in species belonging to the *Firmicutes* phyla, such as *L. reuteri*, *Lactobacillus johnsonii*, *L. acidophillus*, and *L. murinus*. Microbial tryptophan metabolites are low-affinity ligands for AHR; the most potent are indole, skatole, tryptamine, indole-3-pyruvic acid (IPyA), indole-3-acrylic acid (IA), and indole-3-acetamide (IAM), whereas Indole-3-acetic acid (IAA), indole-3-propionate (IPA), indole-3-lactate (ILA), and indole-3-aldehyde (IAID) are the least active [106]. AHR is a ligand-activated transcription factor belonging to the basic helix-loop-helix (bHLH)/Per-ARNT-Sim family. AHRR, a protein that negatively controls AHR-expressed genes by competing with AHR to bind to heterobetaine-responsive elements [107] is downregulated in several tumor types, including colon, breast, gastric, cervical, and ovarian cancers. The knockdown of this gene results in substantial normal breast epithelial cell-independent growth [108]. AHR plays a complicated role in colon cancer. One study showed that AHR expression was raised in tumor tissues from a variety of sources, including the breast, colon, and lung, whereas another study found that AHR deletion led to increased proliferation of colon cells and exacerbated growth in a colon cancer tumor model [109,110,111,112]. The universal oncogene MYC transcription in colon cancer cells first induces the particular profile of AHR in that cancer; AHR then promotes MYC-induced proliferation by turning on the expression of the genes needed for biomass generation. This further improves the uptake of tryptophan and its alteration in colon cancer cells to kynurenine (Kyn), leading to the nuclear translocation of AHR in these cells [112,113]. It is widely known that the Kyn route is in charge of practically all tryptophan metabolism. The three primary enzymes that make up the Kyn pathway are tryptophan 2,3-dioxygenase 2 (TDO2), indoleamine 2,3-dihydrogenase 2 (IDO2), and indoleamine 2,3-dioxygenase 1 (IDO1) [114]. IDO1 and KPM (kynurenine-derived metabolites) have been shown by Bishnupuri et al. to work in concert to stimulate the Phosphoinositide 3-kinase (PI3K)–protein kinase B (AKT) axis, which, in turn, promotes cell proliferation and contributes to the development of cancer. The PI3K-AKT axis subsequently promotes the translocation of β-linked proteins to the nucleus, which increases cell proliferation and resistance to apoptosis [112]. MYC upregulates the production of tryptophan solute membrane carriers, which are converted to Kyn and accumulate in tumor tissues. This is the connection between the MYC pathway, tryptophan metabolism, and CRC [115]. Tryptophan metabolism, particularly indole, IPA, and IA, can also affect the maintenance of the intestinal epithelial barrier in terms of its structure and function through the activation of the pregnancy X receptor (PXR) [116]. PXR influences cancer growth, progression, and resistance to chemotherapy by regulating the expression of the genes associated with proliferation, metastasis, apoptosis, inflammation, and oxidative stress [117].

### 3.5. Bacteriocins

Bacteriocins are large-molecular weight antimicrobial compounds produced by *Lactobacillus acidophilus*, *Bifidobacterium bifidum NCFB 1454*, *Lactobacillus plantarum*, and *Lactococcus lactis*, including microbial peptides and proteins with antimicrobial properties [118,119]. Bacteriocins can be classified into four groups based on their molecular weight. The first category, with molecular weights less than 5 kDa, includes antibiotics such as streptococcal lactic acid, lactate, and mercuric acid [120]. The second category includes heat-stable peptides with molecular weights <10 kDa [121]. The third group consists of thermally unstable peptides with molecular weights >30 kDa and is divided into subclasses IIIa and IIIb, which can disrupt the cell membrane. Proteins containing a lipid or carbohydrate portion make up the fourth group [122]. Bacteriocins have shown anticancer activity by mechanisms including the induction of apoptosis, cell cycle blockade, the inhibition of cell migration, and the disruption of the cell membrane structure [123,124,125,126,127,128]. More specifically, bacteriocins increased cell membrane fluidity, formed ion channels in cancer cell membranes, and increased lactate dehydrogenase release. In addition, bacteriocins promoted the accumulation of intracellular reactive oxygen species, increased the apoptotic index (BAX/bcl2), decreased the expression of Forkhead Box M1 and Matrix metallopeptidase 9, inhibited mitochondrial energy metabolism and glycolysis, reduced their energy supply leading to apoptosis and necrosis or inhibited their migration and proliferation, and ultimately promoted apoptosis and necrosis and inhibited cancer migration and proliferation. Nisin A is a polycyclic antimicrobial peptide that is a member of the class of antibiotics. Nisin A exhibits anticancer activity via restraining tumor cell growth, altering the integrity of cell membranes and pore formation and increasing ion penetration by meddling with phospholipid rearrangements [124]. This antibiotic inhibited tumor cell invasion and the metastasis of different human cancer cells, as well as colon cancer cells [125]. Colicins are high molecular weight bacteriocin (40–80 kDa) produced by *E. coli* [129]. Colistin inhibits tumor cell proliferation and is toxic specifically to cancer cells. For example, colistin E1 exhibited cytotoxic activity against the human colon cancer cell line HT29 [130,131].

### 3.6. Botanical Metabolites

Botanical active ingredients have a unique role in maintaining a healthy gut flora environment. After oral administration, the botanical comes into contact with gut microorganisms in the body, producing metabolites and promoting enterohepatic circulation. Botanical metabolites travel through the digestive tract and are impacted by intestinal microbes and metabolic enzymes when they are secreted into the colon with bile [132]. Specifically, the intestinal flora can metabolize botanical active ingredients alone or co-metabolize them with the host machinery, resulting in metabolites with greater bioavailability and biological activity [132]. Glycoside and saponin are difficult to absorb into the intestinal tract because of their glycosidic bonds and high polarity, resulting in low bioavailability. Biotransformation by the intestinal flora can improve bioavailability. For example, flavonoid glycoside compounds were converted to simple phenolic acids by the intestinal flora through enzymatic degradation, hydrolysis, reduction, dihydroxylation, and other reactions, which were then absorbed, thus improving their bioavailability [133]. Ginsenosides were converted to the more pharmacologically active 20-O-β-D-glucopyranosyl-protopanaxadiol, also known as Compound K, which has an antitumor activity against a variety of cancers, including liver, lung, bladder, colorectal, and gastric cancers, mainly through deglycosylation by the intestinal flora [134,135]. Astragalus, yam, and mulberry polysaccharides were ultimately converted to short-chain fatty acids after glycolysis, which subsequently played a role in antitumor activity, immunomodulation, and energy metabolism [136]. Similarly, these botanical active ingredients influence the composition of the intestinal flora during metabolism and indirectly interfere with tumor development. For example, Wang et al. demonstrated that berberine reduced the abundance of *Akkermansia* (Akk), a mucin degrader associated with intestinal inflammation, in Apc min/+ mice fed a high-fat diet and that excessive mucin degradation reduced the thickness of the intestinal mucus layer, which disrupted the intestinal barrier and ultimately exacerbated colitis and colorectal cancer development. These findings suggest that berberine prevents the onset and progression of CRC by altering the composition of the intestinal microbiota [137,138].

**Table 1 ijms-25-09096-t001:** Gut microorganisms and their metabolites.

Metabolite	Phylum of Microbes	Detection Methods	Antitumor or Tumorigenic Mechanisms	Citation
Butyrate	*Ruminococcaceae*, *Lachnospiraceae*, *Coprococcus comes*, *Faecalibacterium prausnitzii*, *Erysipelotrichaceae*, *Roseburia* spp., *R. intestinalis*, *Butyrivibrio crossotus*, *Clostridiaceae*, and *E. rectale*	FISH microscopy, real-time PCR, 16S rRNA sequencing	Hyperacetylated histones and non-histone proteins and activated GPRs; inhibited NF-κB and Wnt/β-catenin signaling pathways.	[139]
Propionate	*Bacteroides* spp., *Phascolarctobacterium succinatutens*, *Dialister* spp., *Veillonella* spp., *Megasphaera elsdenii*, *Coprococcuscatus*, *Salmonella* spp., *Roseburia inulinivorans*, *Ruminococcus obeum*, *Prevotella ruminicola*, *Alistipes putredinis*, and *Escherichia coli*	Whole genome transcription profiling, real-time PCR, 16S rRNA sequencing	Inhibited HDAC, activated GPR41 and GPR43 receptors, downregulated PRMT1 to induce apoptosis.	[140]
Acetate	*Blautia hydrogenrophica*, *Akkermansia muciniphila*, *Anaplasma* spp., *Prevotella* spp., *Bifidobacterium* spp., *Clostridium* spp., *Rumenococcus* spp., *Streptococcus* spp., *Broutonella* spp., and *Putrefaciens* spp.	16S rRNA sequencing, metagenomic sequencing analyses	Triggered apoptosis or necrosis in CRC cells via the mitochondria.	[54,55,56]
Bile acids	*Bacteroides*, *Eubacterium*, *Bifidobacterium*, *Ruminococcus*, *Peptostreptococcus*, *Propionibacterium*, *Clostridium*, *Lactobacillus*, *Escherichia*, *Streptococcus*, and *Methanobrevibacter*	16S rRNA sequencing	Activated beta-linker protein and Wnt signaling to induce intestinal tumors.	[61,141]
Polyamine	*Campylobacter jejuni*, *Bifidobacterium*, *Bacillus*, enterotoxigenic *Bacteroides fragilis*, and *Enterococcus faecalis*	Metagenomics sequencing	Activated K-RAS leading to phosphorylation of ecotropin-1, which enhanced cellular uptake of polyamines; elevated polyamine levels; promoted the catalysis of eIF5A, which then triggered MYC biosynthesis and promoted CRC.	[96,97,100,142]
Indoles	*Enterobacter aerogenes*, *E. cloacae*, *E. rectale*, *Clostridium. limosum*, *C. tetani*, *C. lentoputrescens*, *C. bifermentans*, and *C. melanomenatum*	16S rRNA sequencing	Activated AHR-och PXR-receptor.	[143,144,145]
IAA	*C. Difficile*, *C. paraputrificum*, *C. lituseburense*, *Bacteroides. ovatus*, *B. thetaiotaomicron*, *B. eggerthii*, *B. fragilis*, *Bifidobacterium pseudolongum*, and *Escherichia coli*	16S rRNA sequencing		
IAID	*Lactobacillus johnsonii*, *L. reuteri*, *L. murinus*, and *L. acidophillus*	16S rRNA sequencing		[146]
IPA	*C. Botulinum*, *C. caloritolerans*, *C. sporogenes*, *C. cadaveris*, *Parabacteroides Anaerobius*, *P. russelli*, and *P. stomatis*	16S rRNA sequencing		
Bacteriocins	*Lactobacillus acidophilus*, *Bifidobacterium bifidum NCFB 1454*, *Lactobacillus plantarum*, and *Lactococcus lactis*	Metagenomics sequencing	Anticancer activity was exerted by altering cell morphology, as well as having a cytotoxic effect.	[147]

## 4. Gut Microbiota Is Involved in Typical Gastrointestinal Cancers

Over ten trillion microbes, including fungi, protozoa, archaea, and viruses, reside in the adult gut [148]. Gut microorganisms and hosts have developed mutually beneficial partnerships through a long period of coevolution where they have chosen each other. Selective colonization by certain microbes is made possible by the stable, nutrient-rich environment of the host. However, these microorganisms and their byproducts also affect the expression of host genes and the control of host lipid metabolism, as well as the growth of the host mucosal immune system, angiogenesis, intestinal epithelium renewal, and the maintenance of intestinal function [149].

### 4.1. CRC

Globally, colorectal cancer (CRC) ranks third in cancer incidence [150]. Global data show a CRC incidence rate of 10% and a mortality rate of 9.4%. As a settler of the gut, the role of the gut microbiota in CRC development cannot be ignored. Abnormal changes in various microorganisms were observed in patients with colon cancer. Among those with elevated abundance were *Bacteroides fragilis*, *Malassezia*, *Aspergillus rambelli*, *Fusobacterium* spp., *Escherichia coli*, *P. micra*, *F. nucleatum*, *Porphyromonas asaccharolytica*, *Akkermansia muciniphila*, and *Desulfovibrio desulfuricans* [151]. Those with reduced abundance included *Bacteroidetes*, *Coprobacter fastidosus*, *Bifidobacterium*, *Butyrivibrio fibrisolvens*, *Clostridium butyricum*, *Roseburia*, *Eubacterium*, *Dorea*, *Coprococcus*, *Faecalibacterium*, *Talaromyces islandicus*, *Sistotremastrum niveocremeum*, *Macrophomina phaseolina*, and *Aspergillus niger* [151]. The gut microbiota–host interactions promote important symbiotic functions; however, in colon cancer, this symbiotic relationship is mainly disrupted by pro-inflammatory, metabolite, and microbial toxin pathways. The colonization of enterotoxigenic *Bacteroides fragilis* (ETBF) results in the creation of *Bacteroides fragilis* toxin (BFT). BFT triggers a cancer-causing signal transmission from the CEC to the mucosal Th17 reaction, resulting in the targeted activation of NF-κB in the CEC of the distal colon, collectively initiating myeloid cell-dependent tumorigenesis in that area (Table 2) [152]. Putative cell wall-binding repeat 2 (PCWBR2), a surface protein on *P. anaerobius*, binds α2/β1 integrins and triggers the PI3K-Akt pathway in CRC tumor cells in response to phosphorylated adhesion plaque kinase. In turn, NF-κB raises cytokine levels to promote a pro-inflammatory response, including IL-10 and Interferon-gamma (IFN-γ) [153]. Colonization with *P. micra* increased the infiltration of Th17 cells and the release of Th17 cytokines (IL-17, IL-22, and IL-23) to promote the formation of CRC in mice. It also upregulated the genes involved in stemness, angiogenesis, cell proliferation, and invasiveness/metastasis. Thus, this triggered CRC development in mice [154]. *Fusobacterium* spp. promoted colon carcinogenesis by discharging the short peptides and SCFAs that recruited bone marrow-derived suppressor cells, thereby inhibiting CD4+ T cell activation [155]. *E. coli* produced colibactin, which induced DNA damage and subsequently promoted dextran sodium sulfate sodium (DSS)-induced CRC development in a mouse model of chronic inflammation and further enhanced the IEC autophagy-deficient mouse model [156].

### 4.2. Gastric Cancer

Similarly, the stomach has a large and diverse microbial community, and the major phyla colonizing the stomach are *Fusobacteria*, *Firmicutes*, *Bacteroidetes*, *Actinobacteria*, and *Proteobacteria* [157]. HP, a member of *Proteobacteria*, is the most dominant and abundant genus in the stomach [158]. HP, a carcinogen closely linked to gastric cancer, primarily influences carcinogenesis via the virulence factors vacuolating cytotoxin A (VacA) and cytotoxin-associated gene A (CagA) [159].

All strains of HP include VacA, a high molecular weight diversely structured protein that endures and is blocked in T cells and macrophages [160]. VacA can be considered as a biomarker associated with gastric cancer [161]. VacA can bind to the receptor proteins tyrosine phosphatase α and β, leading to the vacuolization of gastric epithelial cells, mitochondrial damage, cytochrome C release, and apoptosis [162]. In addition, when VacA is present in the lamina propria, it disrupts the tight connections between epithelial cells and attracts T cells to the infection site. As a result, T cell growth and effector capabilities are impeded, permitting the microorganisms to endure [163]. In addition to altering the host inflammatory response by inhibiting the T cell activation, VacA induces an inflammatory response mediated by NF-κB activation and leading to IL-8 overexpression (Table 2) [164]. Another method by which VacA causes inflammation in the stomach and promotes the development of gastric cancer is by disrupting autophagy [165].

According to Hatakeyama, the strain-specific protein CagA identified stomach cancer (also referred to as microbial oncoprotein) and suppressed the apoptotic pathway in epithelial cells, resulting in morphological abnormalities like cell scatter and elongation, as well as the loss of cell polarity [166,167]. CagA inhibits p53 and RUNX to stop gastric epithelial cells from going through apoptosis [168,169]. Gastric cancer is caused by CagA and T4SS, which trigger the NF-κB pathway, causing DNA damage from reactive oxygen species and inflammation [170]. HP’s CagA contributes to let-7’s epigenetic silencing, which causes Ras to overexpress during the development of stomach cancer [171]. In gastric epithelial cells, Src kinase phosphorylates CagA at the Glu-Pro-Ile-Tyr-Ala (GPIYA) motif. This incident sets off the activation of oncogenesis-promoting pathways like the Ras/Erk, Wnt/β-catenin, PI3K/Akt, Janus kinase (JAK)/signal transducer and activator of transcription 3 (STAT3), and NF-κB (Table 2) [172].

In addition, CagA and VacA have antagonistic effects: CagA activates the nuclear factor of the activated T cells pathway by activating calmodulin phosphatase via phospholipase Cγ, whereas VacA inhibits the nuclear factor of the activated T cells pathway by blocking calmodulin phosphatase activation through the reduction in the VacA-mediated pore-induced calcium inward flow. Furthermore, VacA and CagA exert antagonistic effects on cell morphology [173]. Overall, VacA and CagA substantially contribute to the development of gastric cancer, and their antagonistic effects might influence the development of gastric cancer.

### 4.3. Liver Cancer

The ecological dysregulation of gut microbes usually leads to increased intestinal permeability and the weakening of mucus-associated defenses, which enhances disease susceptibility not only in the local environment of the gut but also in distant organs, especially the liver [174,175]. Anatomically, the liver is connected to the intestine via the portal vein. The liver can receive intestinally derived products (nutrients, microbial metabolites, and microbial components) through the portal vein. Afterwards, these components enter the bile ducts and revert to the intestines from the liver. The enterohepatic cycle continuously exposes the liver to gut-derived elements. The connection between the gut microbes and liver cancer have been demonstrated by many researchers. Compared to healthy controls, the genera of butyrate-producing microbes (*Ruminococcus*, *Clostridium IV*, *Oscillibacter*, *Faecalibacterium*, and *Coprococcus*) were all reduced in early hepatocellular carcinoma, whereas the genera producing lipopolysaccharides (LPSs) (*Klebsiella* and *Haemophilus*) were all increased [176]. Butyrate is used to protect the intestinal mucosa, and LPSs trigger a variety of pathophysiological cascade responses, implying that these microbes promote the development of hepatocellular carcinoma (HCC). Numerous investigations have detailed the gut microbiota linked to HCC in cirrhosis and non-alcoholic fatty liver disease patients. *Erysipelotrichaceae*, *Bacteroides*, *Ruminococcaceae*, and *Enterobacter ludwigii* were substantially increased in patients with HCC, whereas *Leuconostocaceae*, *Fusobacterium*, and *Bifidobacterium* were considerably decreased [90,176,177]. In addition, the metabolites of intestinal microorganisms, such as BAs, indoles, and short-chain fatty acids, play an important role in liver cancer [178]. Of these, BAs are the best known and have even been considered a potential biomarker for liver disease. The relationship between BA metabolism and HCC development has been proven in both animal and human research. Modified levels of BAs lead to resistance to apoptosis and hyperproliferation, which promotes carcinogenesis, in addition to metabolic problems, hepatic inflammation, oxidative stress, and fibrosis [179,180,181]. Additionally, because of the cytolytic effect of BAs, when BAs accumulate in the liver at high concentrations, they might be highly toxic [182,183]. In addition, BAs trigger the ligand-activated transcription factors Fetoprotein receptor (FXR) and Takeda G protein-coupled receptor 5 (TGR5), which regulate the production of bile acids and the process of enterohepatic recycling [184,185]. These receptors consistently influence pathological changes in the precancerous liver, including inflammation [186,187,188], fibrosis, and cirrhosis [189]. The genes regulated by molecular mechanisms include gankyrin, small heterodimer partner, N-myc downstream-regulated gene 2, miR-122, miR-421, and hepatic nuclear factor 1α, an associated protein that triggers ROS production [190,191,192,193,194,195,196]. Hepatic sinusoidal endothelial cells, gallbladder epithelial cells, and Kupffer cells contain a particular bile acid receptor called TGR5 [197,198,199]. TGR5 functions as a negative regulator of HCC, as shown by numerous human investigations, and its anti-inflammatory properties might be the source of its action [200,201,202], enhancement of the barrier function [203], and maintenance of the correct bile acid pool [204].

### 4.4. Pancreatic Cancer

The pancreas is an extragastric digesting organ, and the intestinal flora might impact the development of pancreatic cancer by inducing immunological responses, fostering inflammation, and sustaining cancer-associated inflammation [205,206]. For example, HP infection activates NF-κB and AP-1, resulting in the disruption of biological functions. Elevated levels of IL-8 accelerate inflammation, ultimately leading to pancreatic carcinogenesis [207]. KRAS plays an important role in normal tissue signaling, and mutations in KRAS are present in more than 90% of pancreatic cancer cases [208]. LPSs from HP overstimulate mutations in the KRAS gene and initiate pancreatic carcinogenesis [209,210]. In addition, by upregulating the expression of anti-apoptotic and pro-proliferative proteins such as Bcl-xL, c-Myc, survivin, MCL-1, and the cell cycle protein D1, the persistent activation of STAT3 caused by *Helicobacter pylori* infection advances pancreatic cancer [211,212].

**Table 2 ijms-25-09096-t002:** Gastrointestinal cancer and intestinal microbial metabolites.

Disease	Microbiota	Detection Methods	Disease-Related Microbial Metabolites	Tumorigenic Mechanism of Intestinal Microbes	Citation
CRC	ETBF	PCR gene detection of bft markers	BFT	Activated the NF-kB signaling pathway.	[152]
	*P. anaerobius*	Large-scale fecal shotgun metagenomic sequencing	—	Activated the pro-inflammatory PI3K-Akt pathway.	[153]
	*Fusobacterium* spp.	16S rRNA sequencing	Short peptides, SCFAs	Suppressed bone marrow-derived cells and CD4+ T cells.	[155]
	*E. coli*	Metagenomic sequencing	Colibactin	Induced DNA damage and enhanced autophagy.	[156,213]
GC	*Helicobacter pylori*	Small subunit 16S rDNA clone library	—	Its virulence factor VacA activated the Wnt/β-catenin signaling pathway and the PI3K/Akt signaling pathway, and CagA disrupted the integrity of the connexin, AJ, and TJ.	[157,159]
LC	*Ruminococcus*, *Oscillibacter*, *Faecalibacterium*, *Clostridium IV*, *Coprococcus*, *Klebsiella*, *Haemophilus*, *Erysipelotrichaceae*, *Bacteroides*, *Ruminococcaceae*, and *Enterobacter ludwigii*	16S rRNA sequencing	BA	Activation of FXR and TGR5 receptors.	[176,177]
PC	*Helicobacter pylori*	Enzyme-linked immunosorbent assay	—	Activation of NF-κB, AP-1, KRAS, and STAT3.	[206]

## 5. Immunometabolic Interactions in Gastrointestinal Cancers

The World Health Organization defined immunometabolism as the interaction between inflammation and metabolic diseases [214]. Recently, the concept of immunometabolism has been divided into more specific and distinct parts, namely the contribution of key metabolic pathways to immune cell development, fate, and behavior [215]; changes in the intracellular metabolic pathways in immune cells that alter their functions [216]; and the metabolic reprogramming of immune cells [217]. The principal metabolic pathways involved in immune metabolism are glycolysis, the tricarboxylic acid (TCA) cycle, the pentose phosphate pathway (PPP), fatty acid (FA) oxidation (FAO), FA synthesis, and amino acid (AA) metabolism. Glycolysis is the breakdown of glucose into pyruvate in the cytoplasm under anaerobic conditions, during which two molecules of pyruvate are produced for every molecule of glucose, as well as two molecules of Adenosine 5′-triphosphate (ATP). It is also a source of intermediary molecules in other pathways, including the PPP, AA, and FA metabolic pathways. The TCA cycle is a cyclic process involving the condensation of acetyl coenzyme A with oxaloacetate (OAA) to form citric acid, which is subsequently dehydrogenated, phosphorylated, and ultimately regenerated to oxaloacetate. The PPP pathway involves the metabolism of glucose-6-phosphate to produce NADPH and ribulose-5-phosphate and is characterized by the direct oxidative dehydrogenation and decarboxylation of glucose. FA synthesis is required for the biosynthesis, energy storage, and production of signaling molecules in cell membranes, and its production is tightly dependent on the production of glycolytic, TCA, and PPP intermediates in the glycolysis, TCA, and PPP pathways. The proper development and growth of individual organisms depends on the balance between various metabolic pathways. Gut microbial metabolites can directly or indirectly interact with host immune cells, interfering with immune metabolic processes and leading to changes in the intestinal tumor microenvironment [218].

## 6. SCFAs Were Involved in Energy Metabolism and Immunity through GPRCs and HDACs

The biggest class of receptors in mammals, G protein-coupled receptors (GPCRs), control nearly every aspect of cellular and physiological processes while an organism is alive [219]. Within the GPCR family, GPR41 and GPR43—also referred to as FFAR3 and FFAR2, respectively—are the most significant receptors for SCFAs [220,221]. SCFAs can activate both receptors; the most effective ones are acetate and propionate, which are followed by butyrate and other SCFAs [43,221]. Monocytes, neutrophils, eosinophils, intestinal Tregs, and other immune cells all express FFAR2 [222,223,224]. SCFAs have the ability to block adenylate cyclase, which lowers the amount of ATP produced from cAMP and activates Gi/o proteins through FFAR2 [222]. FFAR2 activation raises the Ca^2+^ concentration, phosphorylates ERK1/2, and activates mitogen-activated protein kinase (MAPK) [225]. ERK activation induces glucose uptake and glycolysis in T cells [226]. By raising the AMP/ATP ratio, SCFAs directly activate Adenosine 5‘-monophosphate-activated protein kinase (AMPK) [227]. AMPK activation in skeletal muscle increases glucose transport and fatty acid oxidation while suppressing the synthesis of proteins and glycogen [226,228]. The key enzymes in gluconeogenesis—phosphoenolpyruvate carboxykinase (PEPCK) and glucose 6-phosphatase (G6Pase)—are expressed less in the liver when AMPK activation occurs [229]. Mammalian target of rapamycin (mTOR), Peroxisome proliferator-activated receptor gamma coactivator-1 alpha, and Forkhead box O3 are just a few of the downstream signaling pathways that AMPK can influence [230,231,232]. Enteroendocrine cells, sympathetic ganglia, K cells, and enteric neurons all contain FFAR3 [233,234]. Through a Gi/O-sensitive route, FFAR3 signaling mediates glucose-stimulated insulin production [235]. Moreover, FFAR3-deficient mice that received the exogenous supplementation of short-chain fatty acids had decreased hepatic fat content and enhanced hepatic metabolism due to the inhibition of hepatic lipid synthesis gene expression [236]. Trompette reported that the alterations in intracellular metabolism brought on by SCFA-activated GPCR increased OXPHOS, mitochondrial mass, and glycolytic capability in CD8+ T cells [237]. SCFAs, particularly butyrate, stimulate the uptake and oxidation of FA in activated CD8+ T lymphocytes. This leads to the TCA cycle being disconnected from glycolytic inputs and favors OXPHOS through FA catabolism and glutamine utilization [238].

HDACs are key to modifying chromosome structure and controlling gene expression. Histone acetylation often promotes DNA dissociation from histone octamers and nucleosome structural relaxation, which, in turn, causes distinct transcription and co-transcription factors to bind to DNA-binding sites and activate gene transcription. Histone acetyltransferase (HAT) transfers the acetyl group of acetyl-coenzyme A to specific lysine residues at the amino-terminal end of histone proteins. HDAC, which deacetylates histone proteins to bind to negatively charged DNA, causing chromatin to curl and inhibit gene transcription, co-regulates histone acetylation and deacetylation in the nucleus. The concentration of SCFAs affects how much of an inhibitory effect they have on HDACs; with higher SCFA concentrations, the inhibition is more noticeable. Of the SCFAs that block HDAC, butyric acid is the most effective [219]. Mice experiments have demonstrated that butyrate may control Treg destiny by increasing histone H3 acetylation at the Foxp3 gene via HDAC inhibition [239]. Additionally, butyrate encourages the acetylation of the Foxp3 protein, which keeps it from being broken down by proteases and increases its stability and function [240]. Butyrate is transformed into butyrate coenzyme A in intestinal epithelial cells. This enzyme passively diffuses into the mitochondria, where it undergoes β-oxidation and fuels the TCA cycle and OXPHOS to provide energy [241]. Natural killer (NK) cells, B cells, T cells, macrophages, and several other immune cells can all have their metabolism impacted by glycolysis and TCA. For instance, it has been demonstrated that activated NK cells, effector T cells, B cells, and LPS-activated macrophages and dendritic cells all exhibit increased glycolysis [242,243,244,245,246], and enhanced glycolysis allows immune cells to produce ATP and other biosynthetic intermediates in sufficient amounts to perform specific effector functions [247]. For macrophages, this includes phagocytosis and the production of inflammatory cytokines (including antigen presentation) and T cells (including the production of effector cytokines such as IL-17 in TH17 cells) [248]. In M1-type macrophages, TCA is destroyed after citrate and succinate production, and the accumulated citrate is excreted from the mitochondria via citrate transporter export, which is then used for fatty acid production [249]. Fatty acid oxidation and synthesis regulate adaptive and innate immunity. For example, M2 macrophages (defined as il-4-activated) rely on fatty acid oxidation programs promoted by the signal transducer and activator of transcription 6 (STAT6) and PPARγ coactivator 1β, which suppress inflammatory signaling [250,251]. Another example is that fatty acid oxidation regulates inflammatory effector and suppressor T cells between Tregs and promotes persistent memory T cells [252]. Unlike fatty acid oxidation, fatty acid synthesis positively regulates the production and function of pro-inflammatory immune cells in the innate and adaptive immune systems. Mitochondrial uncoupling protein 2 promotes the activation of NLRP3 inflammasome by regulating fatty acid synthesis stimulated by Fatty acid synthase, leading to a harmful inflammatory response in sepsis. Fatty acid synthesis is required for the normal differentiation of TH17 cells. A protein called CD5 antigen-like (CD5L) is expressed in so-called “nonpathogenic” TH17 cells, which produce low levels of IL-17 and the anti-inflammatory cytokine IL-10. They play a homeostatic role in the gut, blocking the invasion of gut microbiota and promoting the epithelial barrier function [253].

## 7. Bile Acids Are Involved in Energy Metabolism and Immunity via the FXR

As mentioned earlier, after reaching the ileum, BAs are reabsorbed and biotransformed by microbes into subordinate BAs like DCA and LCA. These derivatives are toxic and carcinogenic but not as harmful as their precursors. The decontamination activity and cytotoxicity of mammalian cells can be enhanced by increased amounts of DCA, potentially leading to inflammation, augmented epithelial apoptosis, and elevated cytokine accumulation [254]. BAs are natural ligands of the orphan nuclear receptor FXR and bind to FXR in the following order: CDCA > DCA > LCA > CA [220]. FXR belongs to the nuclear receptor superfamily and is widely expressed in the gastrointestinal tract. It regulates sugar, lipid, and energy metabolism in addition to acting as a bile acid receptor to control bile acid production [255,256]. In the colorectal tumor environment, the activation of FXR induces genetically altered apoptosis and clearance [257]. This shows that methods for stimulating FXR expression in the colon’s cancerous tumors might be useful for treating the disease. In addition to increasing the expression of glucose transporter 2, which regulates glucose uptake via the FXR-S1PR2-ERK 1/2 signaling cascade, BA-activated FXR in intestinal epithelial cells also lowers cellular energy generation by preventing oxidative phosphorylation [258]. Recent molecular mechanistic studies showed that low levels of FXR were remarkably associated with glycolytic signaling [259]. The overexpression of FXR substantially inhibited the level and capacity of glycolysis in colon cancer cells, resulting in a considerable increase in cellular alkaline and maximal oxygen consumption rates. The overexpression of C/EBPβ substantially accelerated the level and capacity of glycolysis in colon cancer. Therefore, C/EBPβ inhibited the glycolytic capacity of colon cancer cells by suppressing FXR levels. This also suggested that the mechanism by which FXR affected energy metabolism in colon cancer was related to glycolysis, although additional molecular mechanisms are needed for elucidation.

## 8. Tryptophan-IDO1/Kyn/AHR Signaling Pathway and Immune Metabolism

Kynurenine and serotonin are the two main metabolic routes that catabolize tryptophan, the specific mechanisms are shown in Figure 2. Indoleamine-2,3-dioximinogenases (IDOs), notably indoleamine-2,3-dioximinogenase 1 (IDO1), are believed to be significant rate-limiting enzymes in the kynurenine pathway, which has been the subject of research about its immunological function [260]. Nicotinamide adenine dinucleotide (NAD) is a cofactor in the redox reactions that move electrons from one reaction to another, particularly in those that produce energy like the TCA cycle and glycolysis. The kynurenine pathway is essential for the ab initio NAD synthesis [261]. The activation of the IDO1 enzyme triggers the KYN pathway, which, in turn, promotes an increase in the downstream product of the KYN pathway, NAD. This increases the NAD/NADH ratio, leading to the activation of the glycolytic enzyme GAPDH and an increase in the pyruvate product of glycolysis, resulting in elevated levels of acetyl-coenzyme A and enhanced histone acetylation [262]. Histone acetylation levels in LPS/IFN-γ-stimulated macrophages can be increased by glycolytic metabolism, which, in turn, can control IL-1β expression [263]. The stimulation of the KYN pathway by IDO1 activates the AHR, which promotes immunosuppression in the inflammatory microenvironment [264]. AHR activation triggers the production of Tregs, which suppress the inflammatory response [265]. AHR activation in dendritic cells promotes Treg differentiation, thereby contributing to intestinal homeostasis [266]. As a result, IDO1 inhibition leads to a reduction in Treg cell counts, which strengthens the immune system’s response to malignancies [267]. Butyrate, propionate, and BAs increase the conversion of tryptophan to 5-hydroxytryptophan (5-HT) in enterochromaffin cells. Butyrate stimulates 5-HT to create 5-hydroxyindole acetic acid, which binds to Breg cells’ AHR and inhibit it [268].

## 9. Botanical Active Ingredients Affecting Immunometabolic Interactions in Gastrointestinal Cancers

### 9.1. Polyphenols

Dietary polyphenols are organic substances that are naturally present in plants, such as grains, fruits, vegetables, tea, coffee, and wine [269]. Polyphenols comprise a large number of heterogeneous compounds that are usually categorized as flavonoids or non-flavonoids [270]. Most dietary polyphenols have anticancer effects, including against colorectal and gastric cancers (Table 3). It has been demonstrated that the archetypal components of the majority of polyphenols inhibit stomach and colon cancer cells in vitro. For instance, quercetin suppresses the Notch-1 signaling pathway, which lowers the proliferation of colon cancer cells and colon cancer stem cells and selectively generates the KRAS mutations that cause apoptosis in colorectal cancer cells [271,272]. Additionally, quercetin inhibits the AMPKα activators ERK1/2, PKC-δ, and NF-κB, which results in the downregulation of uPA and uPAR expression. This might be linked to the invasion and metastasis of gastric cancer [273]. Apigenin downregulates the Wnt target genes implicated in colon epithelial cell proliferation, such as the cell cycle proteins D1 and c-Myc, via inhibiting the Wnt signaling pathway in colorectal cancer cells in vitro [274]. Additionally, resveratrol controls Wnt/β-catenin signaling and the downstream gene MALAT1, which prevents the invasion and metastasis of CRC cells [275]. The AMPK route plays a crucial role in maintaining cell energy balance, and blocking it diminishes the glycolytic function and migration capacity of CRC cells. The glycolytic agent α-enolase (ENO1) influences the development and spread of CRC through the modulation of the AMPK pathway [276]. Conversely, AMPK controls the glycolytic enzyme pyruvate kinase M2 (PKM2) [277]. Mutations in KRAS in the CRC cells inhibit AMPK phosphorylation via glycolysis [278]. The mutations that activate the Wnt pathway are distinctive features of CRC. Cancer evolution and advancement are impacted by the Wnt/β-catenin signaling route, which interacts with the tumor’s surrounding environment [279]. Tumor-associated macrophages (TAMs) rank among the predominant cellular elements in the tumor’s microenvironment and have been identified as major factors in cancer-related inflammation and the metastasis of tumors. TAMs synthesize the cytokine TGF-β to enhance HIF1α expression, leading to an increase in Tribbles pseudokinase 3 (TRIB3) within cancer cells. Elevated levels of TRIB3 trigger the Wnt/β-catenin signaling route, thereby amplifying the stem cell-like characteristics and cellular incursion in colorectal cancer [280]. Resveratrol also reduces the gastric cancer cell invasion potential by increasing superoxide dismutase activity and decreasing NF-κB transcriptional and heparanase enzyme activities [281]. In mammals, heparanase (HPSE), the sole endo-β-D-glucuronidase, is vital for breaking down extracellular elements and secreting angiogenic and growth-enhancing substances, thus facilitating tumor development, invasion, metastasis, and angiogenesis [282]. Recently, the starvation antitumor effects of HPSE on gastrointestinal cancers have been widely reported. HPSE can enhance the movement and penetration of human gastric cancer cells through the increased phosphorylation of Src kinase and p38 kinase [283]. Src is pivotal in cell communication routes, receiving cues from the preceding PDGFR, EGFR, HER2, Met, and interleukin-6 receptor (IL-6R) and becoming active. Subsequently, the stimulated Src triggers various signaling routes, such as the Ras, MAPK, and STAT3 routes, leading to the activation of the additional elements that facilitate cell growth and differentiation linked to processes like growth, infiltration, metastasis, and angiogenesis [284,285,286]. However, many polyphenols have low bioavailability, indicating that polyphenol metabolites may function in the body [287]. The gut microbiota is crucial in controlling the generation, accessibility, and activity of phenolic metabolites [288]. *Gordonibacter urolithinfaciens* and *Gordonibacter pamelaeae* can biotransform ellagitannins into urolithins [289]. In colon cancer cells, Urolithin A significantly enhanced the expression of cytochrome c and elevated the levels of caspase-3 and caspase-9 in a dose-responsive way, with the stimulation of caspase-3 and caspase-9 in the cytoplasmic lysate potentially triggering apoptotic processes [290]. Bcl-2 functions as a regulator of apoptosis, preventing the movement of cytochrome c from mitochondria into the cytoplasm [291]. P53 induces a halt in the cell cycle through mechanisms that are both reliant and independent [292]. By obstructing Bcl-2 and boosting the production of p53-p21 protein and reactive oxygen species in colorectal cancer cells, Urolithin A is capable of triggering cell cycle arrest and apoptosis [290]. Most microorganisms capable of producing equol are derived from the *Coriobacteriaceae* family, and the plant materials are isoflavone soy glycosides [293]. Equol triggers cell death in stomach cancer cells through a process reliant on mitochondria, akin to the apoptosis triggered by urolithin A in colon cancer cells. This process encourages cell death by altering the mitochondrial membrane potential, elevating cleaved cysteine asparaginase-3 and -9 levels, and reducing Bcl-xL levels. The influence of equol on these specific targets is ascribed to the prolonged stimulation of the ERK1/2 pathway [294]. A different study indicated that equol suppresses the growth of human stomach cancer cells through the modulation of the Akt pathway [295]. Equol triggers Akt dephosphorylation at Thr450, leading to cells experiencing simultaneous cell cycle arrest and apoptosis. Furthermore, equol reduces the levels of cell cycle proteins D1, CDK4, E1, and CDK2, leading to a halt in the G0/G1 phase of the cell cycle in gastric cancer cells. Estragole can demonstrate anticancer properties on stomach cancer cells by removing phosphate groups from Akt at Thr450 and altering cell cycle regulators and the markers of apoptosis. Hesperidin, which is metabolized from flavanones, induces apoptosis and cell cycle arrest via an antitumor mechanism. Hesperidin triggered cell death in colon cancer-afflicted mice by suppressing the ongoing activation of the Aurora-A-driven PI3K/Akt/GSK-3β and mTOR pathways, thereby stimulating autophagy in this particular colon cancer model [296]. Hesperidin also demonstrated the mitochondrial pathway-mediated activation of apoptosis and cell cycle arrest in the 2/M phase at the G site in a mouse colon cancer xenograft model. Hesperidin causes apoptosis in gastric cancer cells by boosting ROS levels and turning on mitochondrial processes [297].

### 9.2. Triterpenoids

Terpenes are metabolites of isopentenyl pyrophosphate oligomers, which are natural plant products with a basic parent nucleus consisting of six polymerized isoprenoid units [298]. Triterpenes are mainly free glycosides such as ginsenosides. The antitumor effects of ginsenosides have been well documented; ginsenoside treatment inhibits tumor proliferation, induces apoptosis and autophagy, and inhibits metastasis and migration; and ginsenosides appear to be promising compounds against gastrointestinal cancers (Table 3) [299,300]. Ginseng is processed by the microbiota in the intestine, mainly by deglycosylation, oxidation, and hydrolysis. Compound K, 20-O-β-D-glu-copyranosyl-protopanaxadiol, is the primary metabolite having pharmacological activity and is obtained through the deglycosylation of ginsenosides Rb1, Rb2, Rb3, Rc, and Rd [134]. The intestinal microbial strains responsible for deglycosylation belong to the phyla Bacteroidetes and Firmicutes, particularly *Bacteroides* and *Lactobacillus* spp., whereas the oxidation and hydrolysis were carried out by the microbes belonging to the phyla *Bacteroides*, *Bifidobacterium*, *Eubacterium*, *Clostridium*, *Lactobacillus*, *Peptostreptococcus*, *Fusobacterium*, and *Prevotella* [301,302,303]. Compound K has greater antitumor properties than those of parental ginsenosides by a fourfold mechanism: (1) overexpressing cAMP-dependent protein kinase, caspase-3, caspase-8, and caspase-9 [206,304]; (2) inhibiting Bcl-2, NF-κB, JAK-1, and STAT3 [305,306,307]; (3) preventing the overexpression of MMP-9 and AP-1 [308], and (4) blocking bFGF-induced angiogenesis [309].

### 9.3. Polysaccharides

Polysaccharides are important biomolecular compounds formed from many monosaccharides linked by bonds [310], and their functions include antioxidant, antitumor, antidiabetic, immunomodulatory, hepatoprotective, hypoglycemic, and gastrointestinal-protective effects [311]. The antitumor effects of polysaccharides on colon cancer are closely related to their chemical composition and conformation [312,313]. A lower molecular weight makes the spatial conformation of polysaccharides susceptible to binding by tumor cells, and high molecular weight polysaccharides might increase spatial site resistance and inhibit their binding to tumor cells. In addition, the structural features of polysaccharides such as β-(1→6) bonds in the main chain can increase the activity of immunoreactive cells or induce apoptosis in tumor cells [314,315]. Therefore, these structural features of polysaccharides might lead to the enhanced inhibition of colon cancer (Table 3). Mechanistically, polysaccharides can regulate the PI3K/Akt, NF-κB, MAPK, and mTOR-TFEB signaling pathways to induce apoptosis and cell cycle arrest, thereby inhibiting the development of CRC [316]. For example, *Scutellaria barbata* polysaccharide (SPS2p) and *Cyclocarya paliurus* polysaccharide (CPP) are involved in the PI3K/Akt pathway in CRC cell lines [316,317]. Apple polysaccharide (AP) extracts inhibit the NF-κB-mediated inflammatory pathway in colorectal cancer [318]. *G. lucidum* polysaccharide (GLP) induces apoptosis in HCT-116 cells by upregulating JNK expression through the MAPK pathway [315]. Lentinan and tea polysaccharides promote autophagy in CRC cells by modulating the endoplasmic reticulum stress (ERS) and mTOR-TFEB signaling pathways [319,320]. The PI3K/Akt, NF-κB, MAPK, and mTOR-TFEB signaling pathways, especially mTOR-TFEB, play key roles in glycolysis, as well as cellular energy metabolism. TFEB performs a part in metabolism, autophagy, energy balance, stress response, and lysosomal biogenesis. A crucial modulator of the autophagy/lysosomal nuclear signaling cascade is TFEB. Moreover, cancer, neurological illnesses, metabolic abnormalities, and lysosomal storage disorders have all been linked to it. mTOR is another major negative regulator. mTOR is a crucial modulator of cell division and growth. The dephosphorylation of TFEB results in its activation and migration from the cytoplasmic lysate to the nucleus when the mTOR is blocked. The activated TFEB in the nucleus stimulates autophagy and autophagosome formation, thereby promoting autophagy by improving the synthesis and function of lysosomes and phagolysosomes, which, in turn, facilitate the breakdown of autophagic substrates [321,322]. It has been demonstrated that tea polysaccharides may act on the mTOR-TFEB signaling pathway and promote cytotoxic autophagy, which, in turn, inhibits the growth of colon cancer cells [320].

### 9.4. Alkaloids

China has long employed berberine (BBR), an isoquinoline alkaloid derived from *Berberis vulgaris*, to treat intestinal infections (Table 3) [323,324]. The bioavailability of BBR is very low owing to its solubility and membrane permeability [325], resulting in the substantial accumulation of the drug in the intestine [326]. However, in the gastrointestinal tract, BBR can be converted to dihydroberberine by nitroreductase from the gut microbiota in the gut wall tissue and then immediately reverted to BBR by oxidation [327]. The mechanism by which BBR inhibits colon cancer involves the regulation of the PI3K/AKT/mTOR pathway and its upstream proteins. mTOR is a major regulator of cellular metabolism, and its dysregulation is strongly associated with cancer. mTOR consists of the mTORC1 and mTORC2 proteins. The PI3K/AKT signaling pathway is a well-established upstream regulator of mTORC1 [328]. BBR is an inhibitor of the PI3K/AKT signaling pathway, which, in turn, inhibits mTOR phosphorylation and promotes cellular autophagy. The promotion of the level of autophagy enhances the arrest of the cell cycle, which, in turn, inhibits colon cancer cells [329].

**Table 3 ijms-25-09096-t003:** Botanical active ingredients and their metabolites were related to gastrointestinal cancer and immune metabolism.

Chemoskeleton	Ingredients	Source	Model	Immunometabolic Interactions	Citation
Polyphenols	Quercetin	Fruits, vegetables, herbs	MGC803, C7901, BGC823, AGS N87, GES-1	Inhibition of AMPK energy metabolism pathway.	[273]
	Apigenin	Parsley, onions, orange, tea, etc.	SW480, HCT15	Inhibition of PKM2 to limit glycolysis.	[274]
	Resveratrol	Polygonum cuspidatum	LoVo, HCT116	Increase in superoxide dismutase activity, decrease in NF-κB transcriptional activity and heparanase activity.	[275]
	Urolithin	Metabolite of polyphenol	HT29, SW480, SW620	Decrease in Bcl-2 and increase in cytochrome c and p53-p21 protein levels.	[290]
	Equol	Metabolite of daidzein	MGC-803	Decrease in Bcl-xL levels and regulation of the Akt pathway.	[294]
	Hesperidin	Citrus fruits	Mice (AOM treatment)	Inhibition of the PI3K/Akt/GSK-3β and mTOR pathways.	[296]
Triterpenoids	Ginsenoside	Ginseng, American ginseng, notoginseng	HT-29, SW480, HCT-116, CT-26	Metabolically deglycosylated, oxidized, and hydrolyzed by intestinal flora to more active compounds.	[134]
Polysaccharides	SPS2p	Whole grass of *Scutellaria barbata*	HT29	Regulation of the PI3K/Akt pathway.	[316]
	CPP	*Cyclocarya paliurus (Batal.) Il jinsk*	SW480	Regulation of the PI3K/Akt pathway.	[316]
	GLP	Ganoderma lucidum	HCT-116	Upregulation of the MAPK pathway.	[318]
	Lentinan	Lentinus edodes	HT-29	Regulation of the mTOR-TFEB signaling pathway.	[319]
	TP	Tea	HCT116	Regulation of the mTOR-TFEB signaling pathway.	[320]
Alkaloids	BBR	Bark of Zanthoxylon clava herculis Linne (Rutaceae)	SW480	Regulation of the PI3K/AKT/mTOR pathway.	[329]

## 10. Botanical Active Ingredients Affecting Immune Cell Interactions in Gastrointestinal Cancers

### 10.1. B Cells

Tumor biology and the response to immunotherapy are directly influenced by the immune and stromal cells that surround the tumor cells, which together make up the tumor microenvironment (TME) [330]. B cells are the primary humoral immunity effector cells in the TME; they are important for the antitumor response and could be novel targets for immune-related therapies [331,332]. It has long been thought that the B cells’ main function is to destroy cancer cells by inducing phagocytosis, which produces antibodies, and antibody-dependent cytotoxicity [333,334]. In addition, by delivering antigens to T cells and encouraging the development of tertiary lymphoid structures (TLSs), B cells perform a variety of roles in initiating immunological responses [335,336]. TLSs are sites of B cell maturation and development; naïve and memory B cells within them stimulate T cells to deliver antigen, which activates T cells to target tumor cells. It has been observed that GC B cells differentiate into memory B cells more readily when they have a low affinity for antigens [337]. In addition, two studies have shown that TLSs have a favorable prognostic effect in patients with CRC [208,338]. A signaling cascade response that upregulates important proteins is made up of STAT3 and the nuclear factor kappa-light-chain enhancer of activated B cells (NF-κB) [339,340]. The crosstalk between these inflammatory signals and the Wnt/β-linker pathway leads to β-linker translocation to the nucleus and the stimulation of the downstream transcription of the oncogenic growth factors cytokine cyclin D1 and c-Myc, leading to inflammation, dysplasia, aberrant crypt foci, and even cancer metastasis [340,341,342]. Anthocyanins inhibit NF-κB phosphorylation and reduce NF-κB nuclear translocation, subsequently leading to the reduced transcriptional activation of inflammatory cytokines [343]. To maintain intestinal homeostasis during inflammatory stress, variable structural domain genes encoding immunoglobulins are upregulated to initiate an immunological reaction. Epigallocatechin gallate (EGCG) regulates a series of ileal immunoglobulins that encode variable structural domains of the heavy chains of immunoglobulin A [344]. Membrane-bound immunoglobulins secrete glycoproteins made by B lymphocytes. Membrane-bound immunoglobulins function as receptors that, when binding to particular antigens, cause B lymphocytes to clonally expand and differentiate into immunoglobulin-secreting plasma cells during the humoral immunity’s recognition phase [345,346]. Chemokines synergistically act with adhesion molecules particular to tissues during plasma cell transport. In particular, B cell homing, development, and function all depend on the chemokine (C-X-C motif) receptor 4 (Cxcr4) [347]. One transmembrane protein that helps move soluble polymeric isoforms of IgA and immunological complexes is called the polymeric immunoglobulin receptor (pIgR) [348,349], and EGCG substantially downregulates pIgR expression in the ileum. The above evidence suggests that EGCG can maintain intestinal homeostasis by regulating immunoglobulins in the ileum. However, whether there is an association between EGCG and gastrointestinal cancer is uncertain. Figure 3 summarizes the signaling of anthocyanins and EGCG affecting changes in B cell metabolism.

### 10.2. Neutrophils

Additionally, EGCG regulates neutrophils, which are significant inflammatory cells and whose infiltration indicates the local inflammatory status of the tumor [350]. In the TME, tumor cells stimulate the neutrophils’ inflammatory characteristics, which cause them to produce neutrophil extracellular traps (NETs) from activated neutrophils [351]. Neutrophils in the TME can promote tumor progression by influencing NET formation, ROS release, and the secretion of protumor chemokines and cytokines [352,353,354]. Interleukin 8 (CXCL8) is a potent chemokine in neutrophils [355]. The TME has a high expression of CXCL8, which encourages colon cancer growth and metastasis [356]. By controlling the PI3K/AKT/NF-κB signaling pathway, CXCL8 overexpression speeds up the epithelial–mesenchymal transition and the malignant phenotype of colon cancer cells [357]. In addition, interleukin 22 activates STAT3 in colon cancer cells to cause the autocrine production of CXCL3, which enhances chemotherapeutic resistance in tumor cells [358]. In colon cancer cells, CXCL8 expression is inhibited by downregulating STAT3 [359]. By reducing the STAT3 expression, EGCG prevents colon cancer cells from proliferating, migrating, and invading [360]. EGCG inhibits NET formation by regulating the STAT3/CXCL8 signaling pathway, which further suppresses the colon cancer cells’ invasion and migration [361]. The specific mechanism by which EGCG inhibited colorectal cancer through the intra-neutrophil signaling pathway was shown in Figure 4.

### 10.3. Macrophages

The primary effector cells of the innate immune system, macrophages, are functionally flexible and can develop into pro- or anti-inflammatory (M1- or M2-like) phenotypes in reaction to various stimuli in the local milieu [362]. M1 macrophages secrete pro-inflammatory cytokines such as TNF-α and interferon IFN-γ, leading to increased mucosal damage and inflammation [363]. On the other hand, M2 macrophages lessen the signs of inflammatory bowel disease by encouraging tissue repair and suppressing inflammatory reactions by overexpressing Arg1, YM-1, and cytokines like IL-10 [364,365]. An imbalance in M1/M2 macrophage polarization has been seen in the mesenteric lymph nodes and spleen in DSS-induced experimental colitis [366]. Upon activation, M1 macrophages express iNOS, M2 macrophages express CD206 and CD163, and macrophages express Tim-1 and TLR4 [367,368]. Many botanical active ingredients are known to limit intestinal inflammation-related diseases by inducing macrophage polarization. For example, quercetin induces macrophages to develop an M2 phenotype with low levels of costimulatory molecules and a phenotype with genes encoding antioxidant proteins, resulting in an increase in the synthesis of the anti-inflammatory molecule IL-10 and a decrease in the secretion of pro-inflammatory cytokines. The effect of quercetin on macrophages is associated with the promotion of HO-1 expression, which stimulates anti-inflammatory effects, M2 polarization, bactericidal capacity, phagolysosome formation, and its downstream molecule CO, which is also involved in mediating microbial scavenging and the generation of anti-inflammatory cytokines by macrophages [273]. Apigenin, a flavonoid, also distorts macrophage polarization, that is, reprogramming in the gut microbiota leading to the secretion of some antiproliferative microbial metabolites at physiological concentrations, which distort macrophage polarization, such as butyrate [369]. However, this hypothesis requires further investigation. By encouraging the buildup of macrophages, which, in turn, produce the proteins essential in maintaining epithelial integrity, such as interleukin-18 binding protein 32 and inhibitors of the Notch signaling cascade, apigenin-induced gut microbiota modification shields animals against colitis [369,370]. Isoglycyrrhizin affects macrophage polarization through the inhibition of the COX-2/PGE2 and IL6/STAT3 pathways; tumors with a severe inflammatory response significantly increase when the COX-2/PGE2 pathway is inhibited due to highly active IL-6 [371,372]. Additionally, the IL6/STAT3 pathway controls the polarization of M2 macrophages and accelerates the development of colitis into cancer [342]. IL-6 stimulates M2 macrophages to produce the pro-tumor cytokine IL-10 [373]; nonetheless, anti-IL-6R monoclonal antibodies stop monocytes from differentiating into M2 macrophages [374]. Berberine also manipulates macrophage polarization and reduces the IL-10 and TGF-β pathways to re-establish its antitumor immune response [375]. In addition, by increasing the expression of MHC-II and CD40 on macrophages, BBR also activates cytotoxic T-lymphocyte (CTL) activity and stimulates the production of CD4+ T cell-derived IFN-γ [375]. By changing the size and shape of macrophages, boosting their phagocytosis and cytotoxicity, and encouraging the generation of cytokines, polysaccharides can stimulate the immune system [376]. Polysaccharides from *Nostoc commune* Vaucher have been shown to activate macrophages in vitro, thereby inhibiting CRC cell lines, as evidenced by increased macrophage size, enhanced phagocytosis, and enhanced TNF-α expression [377]. Ginsenoside Rg1 also regulates macrophage polarization, and its key target has been shown to be the Nogo-B protein [378]. Macrophage migration is hampered by the absence of Nogo-B, which is necessary for macrophage homing [379]. Previous research utilizing Nogo-B^−^/^−^ mice has established the involvement of Nogo-B in inflammatory responses, macrophage infiltration, and macrophage-mediated tissue remodeling in vivo [380]. Nogo-B protein exhibited a dose- and time-dependent inhibition of LPS-induced macrophage activation in in vitro tests [381]. Furthermore, Nogo-B plays a crucial role in the immune response. When overexpressed, it triggers the expression of the pro-inflammatory cytokines CCL-1, TNF-α, IL-1β, and transforming growth factor-β. Furthermore, it improves macrophage migration by drawing macrophages to specific inflammatory sites through chemotaxis [382]. We summarized the mechanisms by which different botanical active ingredients regulated macrophages, as shown in Figure 5.

### 10.4. T Cells

Numerous diverse cell populations, including immune, stromal, and endothelial cells, are found in the TME. These cells secrete soluble signals, such as chemokines, growth factors, or cytokines, which interact with tumor cells to create either favorable or unfavorable microenvironments that control tumor growth and metastasis [383]. Important adaptive immune system mediators, CD4+ T cells, release cytokines that affect other immune cells’ reactions to malignancy [384]. CD8+ cytotoxic T cells (CTLs) are essential for antitumor immunity and are engaged in adaptive immune responses [385]. In addition, CTL depletion promotes immune escape [386]. Figure 6 demonstrated the pathways of action of different plant active ingredients involved in T cell metabolic processes through SCFAs. Litchi procyanidins (LPCs) substantially enrich leukocyte migration and the signaling pathway of T cell receptors across the endothelium, implying that, when CT26 cells are detected by the organism, to get rid of the foreign substance, immune cells congregate at the lesion after crossing the capillary wall [387]. In addition, the potential dependence of LPCs’ immunomodulatory effects on SCFAs boost CD8+ T cell counts [238], reduce the quantity of regulatory T cells in order to reverse immunosuppression [238], and inhibit inflammation in the extraintestinal organs [388,389]. Lactate buildup in the tissue microenvironment is a hallmark of inflammatory illnesses and cancer. Lactate is transformed into citrate in regulatory T cells, which power the tricarboxylic acid cycle [390]. However, citrate might be an injury-associated molecular model extracellularly, promoting the NF-κB translocation to the nucleus by activating NOD-like receptor protein 3 (NLRP3) inflammatory vesicles, which, in turn, drive inflammation [391]. The lactate in CD4+ T helper cells stimulate their differentiation into inflammatory Th17 cells, which result in the buildup of IL-17 [392]. This is connected to the nuclear PKM2/STAT3 pathway [393]. In Apcmin/+ mice, carnosic acid decreased the levels of DL lactate and IL-17, indicating that the inflammatory response in CRC was inhibited.

## 11. Discussion

The intestinal microbiota consists of different species of microbes, archaea, fungi, protozoa, and viruses, with microbes from the phyla *Firmicutes* and *Mycobacteria* dominating the large intestine [394,395]. The gut microbiota substantially affects the host, mainly through the production of vitamins, the metabolism of dietary compounds, the inhibition of expansion, and the systemic penetration of intestinal pathogens [396]. We reviewed the effects of the different metabolites produced by the gut microbiota on the host. The main metabolites include bile acids, short-chain fatty acids, polyamines, and tryptophan, which are involved in the host’s physiological activities, such as nutrition, metabolism, and immunity [397]. We found that the major players in the production of intestinal metabolites are the phyla *Firmicutes* and *Anabaena*, especially *Firmicutes*, which play an important role in the maintenance of homeostasis in the intestinal environment as the major butyric acid producers. Butyrate is a major energy source for colon cells and protects against colorectal cancer and inflammation, regulates gene expression, and has multiple functions, including cell proliferation, apoptosis, and differentiation [398]. In addition, a reduction in butyrate levels might be a key factor in improving the intestinal transit time, as SCFAs can stimulate colonic transit via intraluminal 5-HT release and regulate intestinal motility via the G protein-coupled receptors GPR41 and GPR43 [399]. In addition, it was interesting to note that most SCFAs are derived from the metabolic conversion of dietary fibrous foods by intestinal bacteria, whereas many of the sources of phytoactive constituents that maintain the homeostasis of the intestinal bacterial environment and inhibit gastrointestinal cancers are edible fruits or vegetables. Therefore, a healthy diet seems to be the least costly way to reduce the risk of GI cancer for most people. For example, the consumption of a high-fat diet increases the abundance of sulfate-reducing bacteria [400], which leads to elevated sulfide concentrations and might reduce disulfide bonding in mucus, cleaving the network of polymeric proteins secreted by cuprocytes known as MUC2 (oligomucous mucus gel-forming) [401], leading to defects in the mucus layer and an increase in intestinal inflammation [401], colitis scores [401], and IBD [402]. In contrast, dietary fiber consumption promotes the production of SCFAs (e.g., butyrate and acetate) by the gut microbiota, which exert beneficial effects in maintaining the intestinal barrier’s function and integrity, regulating gene expression, blocking the cell cycle, promoting apoptosis, reducing inflammation, and immunomodulation. Despite the increasing research on gut microbiota, it does not seem to be autonomous and independent in the prevention of gastrointestinal disorders but represents more of an indicator of intestinal health in relation to the intake of adequate fiber. Therefore, maintaining a healthy diet may be the initiating event in the prevention of gastrointestinal cancer.

Colorectal cancer is caused by genetic mutations and microbial interactions. When intestinal microorganisms are imbalanced, pathogenic microbes can cause an intestinal inflammatory response through the activation of recognition receptors, the adsorption and secretion of enterotoxins, or invasion, which further induces oncogene mutations in the intestinal epithelial cells, leading to excessive proliferation of the intestinal epithelium and ultimately the development of tumors. Pathogenic microbes, such as *E. coli*, *Clostridium difficile*, EBFT, *E. faecalis*, *Clostridium difficile*, *Salmonella*, and HP, can promote the development of gastrointestinal tumors through direct or indirect effects. HP can indirectly contribute to gastric cancer by altering the microbial composition of the gastric environment. However, the mechanism underlying this interaction is not fully understood and requires further study.

The immune system plays a complex role in cancer development and has a major effect on colorectal cancer progression [403]. The immune cells involved in the tumor response to colon cancer include B cells, T cells, macrophages, mast cells, and neutral leukocytes. We found that the immune system has a dual role in gastrointestinal cancer. For example, macrophages are polarized to the M1 type, which is pro-inflammatory and promotes tumor progression, and polarized to the M2 type, which is anti-inflammatory and hinders tumor progression. Moreover, the anti-sulfated glycosaminoglycan antibody produced by B cells has an antitumor effect on colon cancer, and B cells can also be activated to inhibit LM-CRC, illustrating that the role of immune cells in cancer is extremely complex and that understanding the role of immune cells in colon cancer plays an important role in clinical treatment.

The therapeutic role of botanical active ingredients in gastrointestinal cancer cannot be ignored, because flavonoids such as quercetin, apigenin, and anthocyanins can intervene in gastrointestinal cancer. The role of flavonoids in preventing gastrointestinal cancer might be related to their bioavailability. However, the exact mechanism remains unclear. Resolving this problem may greatly contribute to the development of flavonoids with medicinal value. In addition, the other active ingredients of traditional Chinese medicine, such as alkaloids, triterpenoids, ginsenosides, and polysaccharides, play a therapeutic role by regulating the intestinal environment or immune cells. This review has detailed the specific ways in which various types of ingredients are involved in the immune mechanism and thus treat gastrointestinal cancers. However, it is not clear how they inhibit gastrointestinal cancers through the intestinal flora. It is clear that botanical active ingredients almost always produce an interventional effect on gastrointestinal cancer by changing the composition of the intestinal flora, increasing the number of beneficial microbes, and decreasing the number of pathogenic microbes. Therefore, clarifying how gut microbes affect physiological and pathological changes in the intestinal tract is crucial for understanding the specific mechanisms by which the botanical active ingredients inhibit gastrointestinal cancers.

## 12. Conclusions

This review aimed to elucidate the composition of the intestinal flora and their metabolites in gastrointestinal cancers and to determine how the flora metabolites influence the progression of gastrointestinal cancers in terms of the immune system. Our findings suggest that the gut microbiota is a complex biological system. Gut flora, intestinal metabolites, and immune cells affect gastrointestinal cancer and crosstalk with each other. Although metabolomics technology is rapidly advancing, it is not sufficient to fully explore the contribution of metabolites to communication between the gut environment and host diseases. Therefore, a concerted effort is necessary to understand the relationship between the gut microbiota, metabolites, and the host’s gut health. This contributes to the improvement of human health.

## Figures and Tables

**Figure 1 ijms-25-09096-f001:**
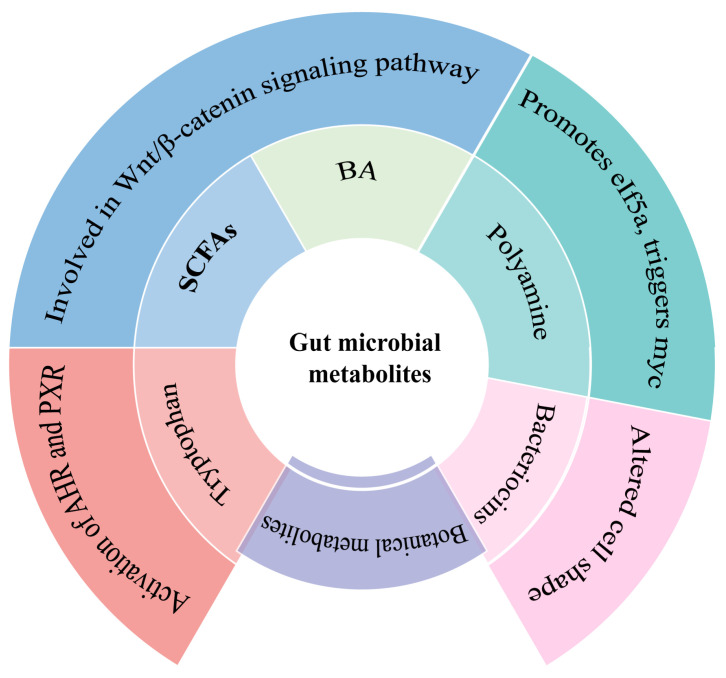
Intestinal metabolites and mechanisms.

**Figure 2 ijms-25-09096-f002:**
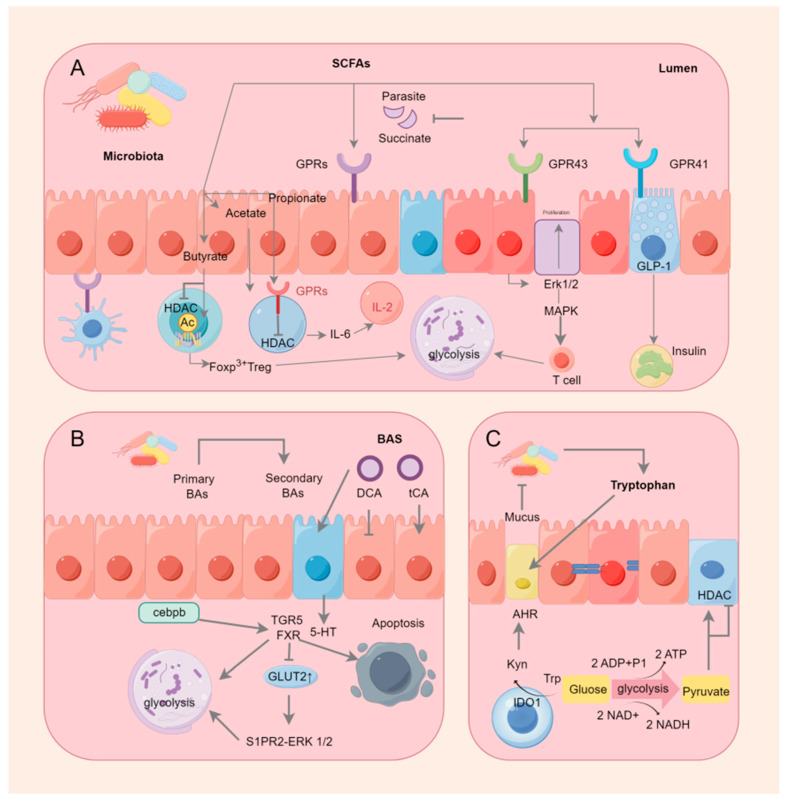
Gastrointestinal microorganisms and immune metabolism.

**Figure 3 ijms-25-09096-f003:**
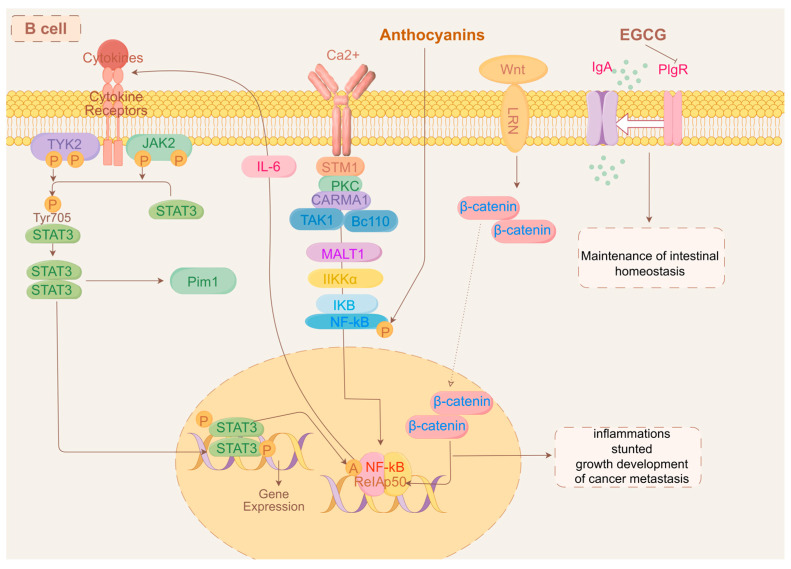
Mechanism of action of anthocyanins, EGCG, and B cells.

**Figure 4 ijms-25-09096-f004:**
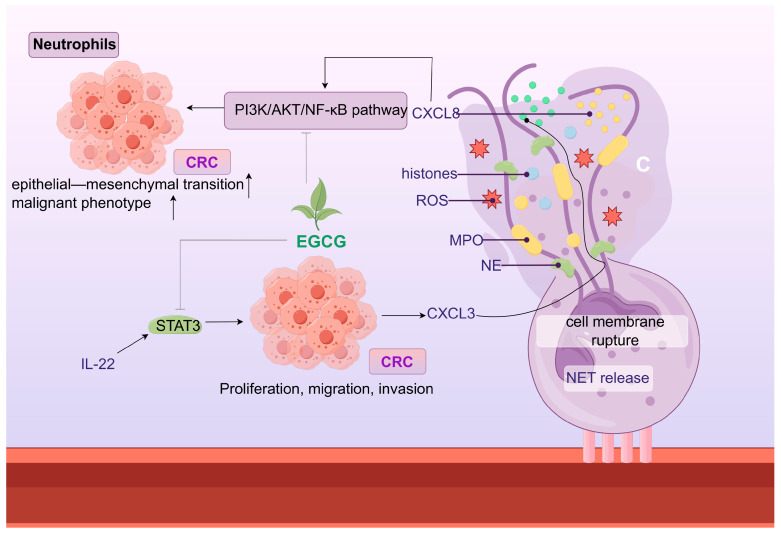
EGCG inhibited CRC through intra-neutrophil signaling.

**Figure 5 ijms-25-09096-f005:**
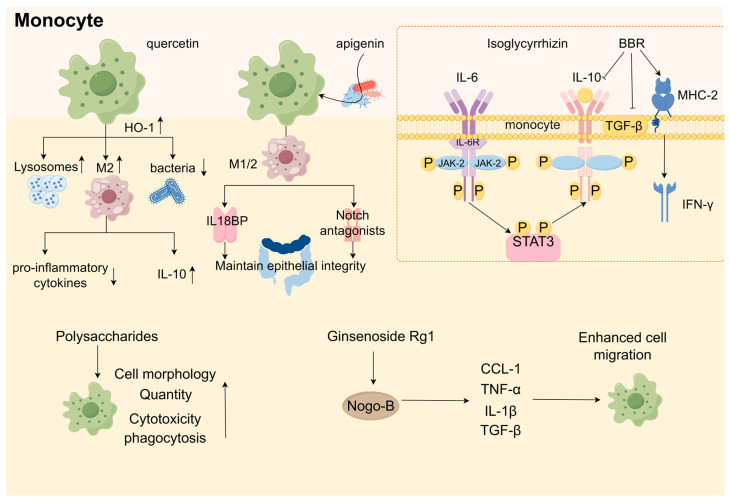
Mechanisms of action of botanical active ingredients on macrophages.

**Figure 6 ijms-25-09096-f006:**
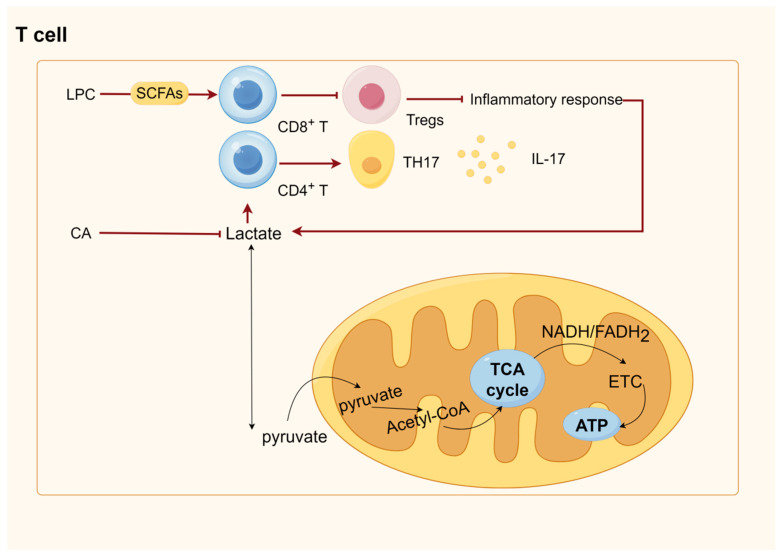
Botanical active ingredients act on T cells via SCFAs.

## Data Availability

No data were used for the research described in this article.

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
