# Peer review of "A Narrative Review: Immunometabolic Interactions of Host–Gut Microbiota and Botanical Active Ingredients in Gastrointestinal Cancers"

_ijms, 2024, doi:10.3390/ijms25169096_

Round 1

Reviewer 1 Report

Comments and Suggestions for Authors

I appreciate the detailed exploration of the impact of the gut microbiota on gastrointestinal cancers in your review. However, to improve the rigor and clarity of your narrative review, I suggest the following revisions:

1.Also in a narrative review, it is essential to outline the search criteria and methodology used to collect the included studies. This includes stating the databases consulted, keywords used, inclusion and exclusion criteria, and any date limitations applied.

2.SANRA (Scale for the Assessment of Narrative Review Articles) guidelines must be followed to ensure the quality of the narrative review. This includes clearly stating the objective of the review, presenting a structured and logical narrative, critically evaluating sources and their relevance to the topic, including a summary of the evidence and its implications, ensuring transparency in the selection and presentation of the literature, and highlighting the strengths and limitations of the review. Also, specify in the title that this is a narrative review.

3.The role of the microbiota in gastrointestinal health and disease appears nuanced. It is critical to delineate whether the microbiota directly affects gastrointestinal cancers or serves as an indicator of dietary habits (e.g., fiber and plant-based foods) and overall health. Discussing whether the role of the microbiota is an independent factor or simply reflective of a healthy diet will provide more clarity.

4.The connection between gut microbiota and gastrointestinal cancers remains complex. Although some microbial metabolites have been shown to impact cancer development and progression, it is essential to discuss whether these effects are direct or secondary to dietary patterns that promote a healthy microbiome. Exploration of this issue may help readers understand whether a more effective strategy is to intervene in the microbiota itself or to promote changes in diet.

Comments on the Quality of English Language

English is fine

Reviewer 2 Report

Comments and Suggestions for Authors

·         The abstract is very weak in its current form. Please include the following: (1) define the process to literature searching (e.g., PubMed and Scopus were searched...etc.), (2) clearly state the objective of the review and succinctly summarize the key findings, (3) connect the findings to practical implications and the potential impact on gut health.

·         The terms “herbal compounds” is not appropriate as used in this paper. Please find another suitable term. For example, quercetin is not a herbal compound.  

·         In introduction, clearly state the research gap and the purpose of the review. Why herbal compounds would be benefit in gastrointestinal cancers? I would suggest to present the aim of the review with regards to what is currently known by in previous/recent reviews, therefore highlighting the added value of this review. 

·         Although this is a narrative review, a method section must be included. The section should include search terms, study types (e.g., in vivo, in vitro…etc.), inclusion and exclusion criteria, databases and a period of time encompassed by the search.

·         Some studies contain little synthesis of the information. Please provide more information. For example, a higher butyrate supply lowers the risk of colon cancers caused by carcinogens, in part by inducing apoptosis, How? Butyrate inhibited the proliferation of colon cancer cell lines and induced cell cycle arrest via multiple molecular pathways, What metabolic pathways? GPR accelerated the development of colonic inflammation and CRC in several experimental model systems, How?

·         In tables 1-2, I would also suggest adding the methodology that was used in the individual cited publications. I would like to point out that the microbiome assessment is largely dependent on the choice of testing method, e.g. 16 SRNA, metagenomics sequencing, and others.

·         Please define abbreviations in the first used (e.g., NF-κB).

·         Please used italics for genus name only (e.g., Bifidobacterium, Veillonella, Salmonella).

·         Please add more columns to clarify table 3.

·         I would suggest adding tables to clarify sections 2.2-2.6.

·         Please check all abbreviations- many are not listed at the end.  

·         Need to follow the journal guidelines for references.

Comments on the Quality of English Language

Please consult with an English language speaker or MDPI Author Service to edit the whole manuscript.

Round 2

Reviewer 1 Report

Comments and Suggestions for Authors

The authors have fixed several shortcomings and added the SANRA guidelines to the paper.I suggest to emphasize both in the abstract and in the discussion that, despite the extensive research and the growing number of studies on the microbiota, it seems to represent more an indicator of intestinal health status related to adequate fiber intake rather than an autonomous causal agent in the prevention of gastrointestinal diseases. This emphasis will help clarify the role of the microbiota in a more balanced context, reflecting the evidence that closely links it to dietary habits, particularly fiber intake.

Comments on the Quality of English Language

English is fine 

Reviewer 2 Report

Comments and Suggestions for Authors

No further comments.

Comments on the Quality of English Language

Minor language editing is required.

Author Response

Point-by-point response to referee Reviewer #2:
Thank you for your letter and for the reviewers’ comments concerning our manuscript
entitled “Immunometabolic interactions of host-gut gut microbiota and herbal
compounds in gastrointestinal cancers” (ID: ijms-3152044).Those comments are all
valuable and very helpful for revising and improving our paper, as well as the
important guiding significance to our researches. We have studied comments carefully
and have made correction which we hope meet with approval. To make it easier for
you to read, we've marked the changes in red.
Reviewer’s comment 1:
Minor language editing is required.
Author’s response 1:
Thanks for your suggestion. However, we do invite a friend of us who is a native
English speaker from the USA to help polish our article.For example, we changed
controlled to controlled on page 7, line 9. e.g. the first sentence of the Introduction
section on page 1, “Gastrointestinal cancers encompassed a broad spectrum of
diseases involving several organs, including the esophagus, stomach, liver, pancreas,
colon, and rectum, resulting in an enormous global health burden” was optimized to
“Gastrointestinal cancers covered a wide range of diseases that affect various organs
such as the esophagus, stomach, liver, pancreas, colon, and rectum, causing a
significant health burden worldwide”. Another example is that the first sentence
“Short-chain fatty acids (SCFAs) were primary final products of gut microbial
metabolism in the human colon and were mainly derived from substrates of dietary
carbohydrates (non-starch polysaccharides, oligosaccharides, lignans, and similar
polysaccharides) that were anaerobically fermented by intestinal microorganisms” in
the section 3.1 on page 3 was optimized to “Short-chain fatty acids (SCFAs) were the
main products of gut microbial metabolism in the human colon, primarily coming
from dietary carbohydrates like non-starch polysaccharides, oligosaccharides, lignans,
and similar polysaccharides that were fermented by intestinal microorganisms without
oxygen”. And we hope the revised manuscript could be acceptable for you.

Round 3

Reviewer 1 Report

Comments and Suggestions for Authors

The authors has fied the paper following my suggestions

Comments on the Quality of English Language

English is fine